# Lexical innovations are rarely passed on during one's lifetime: Epidemiological perspectives on estimating the basic reproductive ratio of words

Andreas Baumann [1,2]*

1 Department of German Studies, University of Vienna, Vienna, Austria, 2 Research Network Data Science, University of Vienna, Vienna, Austria

* andreas.baumann@univie.ac.at

## Abstract

Lexical dynamics, just as epidemiological dynamics, represent spreading phenomena. In both domains, constituents (words, pathogens) are transmitted within populations of individuals. In linguistics, such dynamics have been modeled by drawing on mathematical models originating from epidemiology. The basic reproductive ratio is a quantity that figures centrally in epidemiological research but not so much in linguistics. It is defined as the average number of individuals that acquire a constituent (infectious pathogen) from a single individual carrying it. In this contribution, we examine a set of lexical innovations, i.e., words that have spread recently, in four different languages (English, German, Spanish, and Italian). We use and compare different ways of estimating the basic reproductive ratio in the lexical domain. Our results show that the basic reproductive ratio can be somewhat reliably estimated by exploiting estimates of lexical age of acquisition and prevalence but that the derivation based on diachronic corpus data comes with certain challenges. Based on our empirical results, we argue that the basic reproductive ratio can inform about the stability of newly emerging words and about how often such words are successfully propagated in linguistic contact events. Our analysis shows that an average lexical innovation that has spread in the previous two centuries has been passed on by each individual only to a handful of contacts.

## 1. Introduction

Language change and epidemiological dynamics share some commonalities. This insight is not particularly new: already in the 1980s, quantitative laws of linguistic change have been inspired by epidemiological considerations [1, 2] and epidemiological models were transferred to the domain of cultural evolution [3]. Conceptually, what both domains have in common is that constituents, be they words (or other parts of linguistic knowledge) or pathogens, are transmitted within populations of individuals. Both, language change and epidemiological dynamics represent spreading phenomena [4–8].

**Data Availability Statement:** Preprocessed data and code relevant to reproduce the present analysis are available as a GitLab project at https://gitlab.com/andreas.baumann/basic_reproductive_

ratio. All estimates are collected together through the script collect_estimates.R in a single dataframe saved as estimates/estimates_all.csv. The same dataset can be downloaded from the long-term data repository of the University of Vienna at https://phaidra.univie.ac.at/o:2072321. Data and code are licensed as CC BY-NC https://creativecommons.org/licenses/by-nc/4.0/.

**Funding:** This study was financially supported by University of Vienna in the form of open access funding received by AB. No additional external funding was received for this study.

**Competing interests:** The authors have declared that no competing interests exist.

This conceptual similarity has led to fruitful methodological transfer. So, logistic (S-shaped) dynamics, that are abundantly employed in epidemiology, function as a standard model in the study of language change [9–15]; and on a more general level, concepts from evolutionary theory (mutation, selection, drift) found various applications in the study of language change and evolution [12, 16–20].

Interestingly, one concept that is central to the field of epidemiology, the basic reproductive ratio, has received less attention in the study of language than it might deserve. In epidemiology, the basic reproductive ratio (or also: basic reproduction number, basic reproductive number), in short $R_0$ ('R nought'), is defined as the expected number of individuals in a population of individuals susceptible to a disease that are infected by a single infected individual [21–24]. That is, it measures how many secondary infections there are per newly introduced infected person. A central observation is that if $R_0 > 1$ then a pathogen can spread, while it will, in the long run, go extinct, if $R_0 < 1$.

The basic reproductive ratio was explicitly applied to and formally derived for the lexical domain by Nowak and colleagues in their contributions in the *Journal of Theoretical Biology* [25]; with surprisingly low reception in the field of linguistics) and *Nature* [26], and later by Solé and colleagues [27], as well as in research on phonological change of our own [28]. In the linguistic domain, $R_0$ measures the expected number of individuals that learn a newly introduced linguistic constituent (like a new word) from a single individual that already knows and uses that constituent, and the result about the consequences of $R_0$ being greater or less than one applies–*mutatis mutandis*–as above.

However, theoretical epidemiology offers more than that. First, it yields a threshold for herd immunity, i.e., a minimum fraction of individuals that cannot pass on a disease so that the disease cannot spread even if there are susceptible individuals [29–32]. This concept has a linguistic analogue. The herd immunity threshold can be interpreted as a minimum fraction of speakers resisting some linguistic change which is necessary for that change not to be successful in the end. Put differently, it can help to address the following question: how many speakers really need to reject, say, a certain word in order for the word to fall out of linguistic usage in the whole population?

Second, and more fundamentally, the basic reproductive ratio, since it is defined as the expected number of successful secondary infections per primary infection during the whole infectious period, implicitly informs about contact dynamics. Again, the matter is relevant for the linguistic domain as well. If we know about the average size of personal contact networks, $R_0$ informs us about the fraction of contacts in that network that really adopt a given linguistic innovation, such as a new word, from a single individual during the period of usage of that word. The latter restriction is crucial. It entails that $R_0$ is dimensionless in the sense that the quantity is always put into relation with the duration of the period during which the innovation can be passed on in the first place. Put differently, $R_0$ is not a rate and it does not tell us anything about the timescale of spreading events, nor does it inform us about how often such innovations are actuated in the first place.

Obviously, one needs empirical estimates of $R_0$ to derive such information. Theorems about how $R_0$ can be estimated belong to the standard results of theoretical epidemiology. These results relate $R_0$ to epidemiological and ecological concepts, in particular (i) age of first infection, (ii) equilibrium prevalence, and (iii) growth data [22, 23, 33, 34]. The aim of this contribution is to transfer these results to the study of language and to test them empirically in the lexical domain by estimating the basic reproductive ratio of words based on a simple model of the spread of linguistic constituents. It will be shown how the basic reproductive ratio of words relates to lexical prevalence data, age-of-acquisition ratings, and diachronic trajectories.

The paper is structured as follows. In the following two subsections, we will first review fundamental epidemiological concepts and standard results (section 1.1), and then transfer them to the linguistic domain (section 1.2). After that, we will test predictions against diverse diachronic and synchronic linguistic data from four languages (English, Spanish, German, Italian; sections 2–3). In particular, we will see that deriving the basic reproductive ratio from diachronic data does not come without problems. Mindful of the limitations that the adopted model and the empirical approach have (section 4.1), we will relate our findings (a) to the stability of lexical constituents, (b) to the role that individuals and their personal networks play in lexical diffusion, and (c) to the relationship between linguistic prevalence, acquisition and change (section 4.2). In doing so, we do not test the analyzed model directly. Rather, the model functions as a vehicle from which different estimation methods are derived. In that sense, the underlying model is only tested indirectly by comparing its predictions against observed data. Based on our empirical analysis, we will conclude that an average newly emerging word is passed on by each individual only rarely, i.e., to a relatively small number of contacts not yet knowing the word that the individual encounters during their lifetime.

## 1.1 Key concepts and results in mathematical epidemiology

This section provides a very brief overview of the formal derivation of the basic reproductive ratio in mathematical epidemiology. For an excellent and certainly more exhaustive review see Heffernan et al. [23] (2005) (as well as [35] for criticism). Epidemiologists are interested in the growth of the number of infected individuals within a population. Let us assume that we can divide the whole population into several pairwise disjoint subpopulations, or compartments, some of which consist of infected individuals. These compartments can reflect, e.g., social structure, age structure, or stages of a disease. Assuming that there are $n$ compartments, let $x \geq 0$ be a $n$-dimensional vector that lists the number of individuals in each of the compartments. Let $I$ and $S$ denote the set of infected and not yet infected (susceptible) compartments, respectively.

In what follows, we describe a way of deriving the basic reproductive ratio referred to as the 'next generation method' [36]. Let $F_i(x)$ denote the rate of new infections that enter an infected compartment $i \in I$ and let $T_i(x)$ measure the rate of net-transitions that go out of compartment $i$. The rate of change of $x_i$ then is given by $\dot{x}_i = F_i(x) - T_i(x)$.

Given all rates $F_i(x)$ and $T_i(x)$, we can compute the 'next generation matrix' $FT^{-1}$, where $F = \left[\frac{\partial F_i(x_0)}{\partial x_j}\right]_{i,j \in I}$ and $T = \left[\frac{\partial T_i(x_0)}{\partial x_j}\right]_{i,j \in I}$. Here, $x_0$ is the population dynamic equilibrium if there is no disease. The basic reproductive ratio then is given by the dominant eigenvalue $\rho$ of the next generation matrix, i.e., $R_0 = \rho(FT^{-1})$. The intuition behind this definition is that reproduction depends (i) on the number of newly infected individuals and (ii) on how long individuals are infectious. The former is captured by $F$. The latter is captured by $T$, which collects all net-transition rates. Then, $T^{-1}$ can be interpreted as measuring all waiting times before an individual exits all infected compartments.

The derivation becomes easy and intuitively clear if there is only a single infected compartment. In the 'susceptible-infectious-susceptible' (short: SIS) model of infectious diseases the population is split into two compartments, susceptible and infected individuals [32]; for a review of the basic reproductive ratio in other model setups see van den Driessche (2017). In that case, $S$ and $I$ consist of one compartment each. Individuals can become infected or recover, in which case they switch back to the susceptible compartment. With a slight abuse of notation, let $S$ and $I$ denote the respective fractions of individuals. We assume that both subpopulations are homogeneously mixed. Then, SIS-dynamics can be formulated by means of a

pair of ordinary differential equations

$$
\begin{aligned}
\dot{S} &= -\beta SI + \gamma I - \mu S + \mu \\
\dot{I} &= \beta SI - \gamma I - \mu I
\end{aligned}
\tag{1}
$$

where we assume $I+S = 1$, i.e., overall population size is normalized and kept constant. Here, $\beta$ is the (per capita) rate of disease transmission and $\gamma$ is the rate at which infected individuals recover. The demographic parameter $\mu$ denotes mortality rate (i.e., the duration $1/\mu$ corresponds to one generation). Note that the term $+\mu$ in the first equation ensures that population size is constant. The disease-free equilibrium is $(S,I) = (1,0)$, so that $F = \beta S|_{S=1} = \beta$. The transition rate is $T = \gamma+\mu$ and the inverse $T^{-1} = 1(\gamma+\mu)$ is the expected waiting time until recovery takes place. Hence, $R_0 = \beta/(\gamma+\mu)$.

There are several ways of estimating the basic reproductive ratio empirically [23, 37, 38]. To begin with, it could be determined by estimating the model parameters it is defined by. Clearly, in the SIS model, estimates of $\beta,\gamma$, and $\mu$ suffice to determine $R_0$. In practical terms, however, it is not always easy to obtain reliable estimates for $\beta$ and $\gamma$, and this more generally holds true for more complex models [23]. There are three other interesting ways for obtaining $R_0$ estimates.

First, $R_0$ can be estimated based on the prevalence of the disease at its population-dynamic equilibrium [32]. The argument goes like this: Let $p_I$ denote prevalence, i.e., the fraction of all infected individuals. Then for an infected individual, $1-p_I$ is the probability of interacting with a susceptible individual, so that the number of infection events induced by that individual is $R_0(1-p_I)$. At population-dynamic equilibrium when there is no change in the number of infected individuals, however, this number is exactly one and hence $R_0 = 1/(1-p_I)$.

We can retrace this derivation in the SIS-model by computing prevalence $p_I = I^*$ (the star denoting a population-dynamic equilibrium). Thus, setting $\dot{I} = 0$ we find that $p_I = I^* = 1 - \frac{\gamma+\mu}{\beta} = 1 - 1/R_0$, which implies the above formula.

Second, we can use age of first infection $AoI$ to estimate $R_0$ [22, 38]. Assume that in the population, all ages are homogeneously mixed and equally frequent. That is, the population follows a rectangular age distribution. Let $LE$ denote expected lifetime (life expectancy). If we assume that, once infected, individuals cannot become susceptible again, then at population-dynamics equilibrium, $AoI/LE$ individuals are not yet infected, and $p_I = 1-AoI/LE$ are already infected. The above considerations about equilibrium prevalence entail that $R_0 = LE/AoI$.

Third, the basic reproductive ratio can be estimated based on the intrinsic (exponential) growth rate $r$ of a pathogen [23, 39]. In the early phase of an outbreak, intrinsic growth rate is implicitly given by $\dot{I} = rI$. That is, given a trajectory of prevalence, one can estimate $r$. The basic reproductive ratio then is $R_0 = 1+rL$, where $L$ is the duration of the infected period.

Irrespective of how the basic reproductive ratio is estimated, the herd immunity threshold can be computed as $H = \frac{R_0 - 1}{R_0} = 1 - 1/R_0$, which turns out to equal equilibrium prevalence [29–32]. If the fraction of individuals in the population that cannot transmit the disease exceeds this threshold, the disease cannot spread and the fraction of infected individuals will approach zero.

## 1.2 Transferring concepts and results to linguistics

The epidemiological concepts and results outlined above can be transferred to the linguistic domain. Here, in place of pathogens, there are linguistic constituents like words, phonemes, or constructions that are passed from one individual to another [4, 6, 8, 20]. Crucially, this is done through multiple interactions among individuals. Instead of infected and susceptible

individuals, let us divide the speaker population into heterogeneously mixed users and non-users of a linguistic constituent [25, 28, 40]. Users already know and use the constituent of interest while non-users still have to acquire it. Let us define the respective fractions as $U$ (for user) and $N$ (for non-user). We can then define the system

$$
\begin{aligned}
\dot{N} &= -\beta UN + \gamma U - \mu N + \mu \\
\dot{U} &= \beta UN - (\mu + \gamma)U.
\end{aligned}
\tag{2}
$$

Here, $\beta$ defines the per capita rate at which the constituent is transmitted from a user to a non-user. Let us refer to such events as instances of *propagation*. Upon successful propagation, the non-user switches to the user class. Note that $\beta$ measures propagation rate under the condition that users and non-users interact, and not that a user utters the constituent. In fact, Nowak et al. (2000) [26] define this rate as $\beta = bq\phi$, where $b$ is the number of contact events, i.e., linguistic interactions, in which learning can take place, $q$ is the probability to adopt a constituent if encountered once, and $\phi$ is relative utterance frequency of the constituent. Hence, $\beta$ can expected to be a function of utterance frequency, learnability, and the number of contact events. Note that the number of contact events can be much higher in linguistic propagation than in the transmission of pathogens since linguistic communication is restricted to physical contacts to a lesser extent. Due to the assumption of heterogenous mixing, the rate at which non-users switch to the user class is the product of propagation rate, the fraction of users, and the fraction of non-users (mass-action transmission; [23, 35]). The parameter $\gamma$ defines the rate at which users stop using the constituent (for instance, because they forget it). It does not depend on contacts among individuals. As above, $\mu$ defines per capita mortality rate. Note that we assume all users subject to death to be added to the non-user class. This represents that new-born individuals are not yet equipped with the linguistic constituent. Apart from $\mu$, age structure is not reflected in the model. In particular, heterogeneous mixing implies that interaction events are not conditioned by age. Propagation can occur in interactions with young individuals (reflecting the scenario of L1 acquisition in children that interact with their caretakers) as well as in interactions among old individuals (reflecting the adoption of words in everyday speech). All model parameters are summarized in Table 1. The model is schematically represented in Fig 1 (left).

As above, the basic reproductive ratio of this system is $R_0 = \beta/(\gamma+\mu)$, and if $R_0 > 1$ then it asymptotically approaches a stable population-dynamic equilibrium with $U^* = 1 - \frac{\gamma+\mu}{\beta}$. That is, the basic reproductive ratio and the number of users at equilibrium increase as propagation rate increases, and as the rate of abandoning a constituent and mortality rate decrease.

The ODE above models logistic, i.e., sigmoid, growth. Since $N+U = 1$, the system can be reformulated as

$$
\dot{U} = U(\beta - \gamma - \mu)\left(1 - \frac{1}{U^*}\right).
\tag{3}
$$

**Table 1. Epidemiological and linguistic model parameters.**

| Parameter | Epidemiological reading | Linguistic reading |
|---|---|---|
| $\beta$ | Rate of successful disease transmission | Rate of successful propagation of a constituent, depending on the average number of contacts, learnability, and utterance frequency |
| $\gamma$ | Rate of recovery | Rate of abandoning a constituent |
| $\mu$ | Mortality rate | Mortality rate |

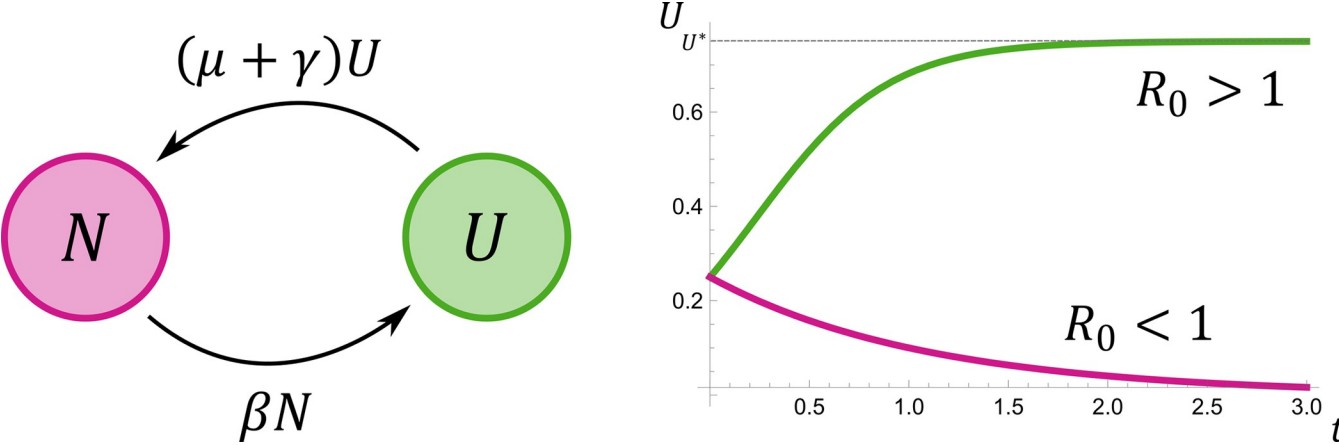

**Fig 1. Compartment model.** Left: schematic representation of the compartment model. Right: trajectories of logistic growth (green) and decline (magenta). Parameters: $U(0) = 0.25$ and $\gamma + \mu = 1$ in both cases; $\beta = 4$ and $\beta = 0.1$ for the growing (green) and declining (magenta) case, respectively.

It can be solved analytically to yield

$$U(t) = \frac{U^*}{1 + \left(\frac{U^*}{U_0} - 1\right)e^{-rt}} \tag{4}$$

where $U_0 = U(t) > 0$ is the initial fraction of users [40]. Here, the intrinsic growth rate is given by $r := \beta - \gamma - \mu$, and $U^* = \lim_{t\to\infty} U(t) = 1 - \frac{\gamma + \mu}{\beta}$ defines the carrying capacity of the system. The logistic equation was used abundantly to model sigmoid (S-shaped) growth in linguistics on various levels of linguistic analysis [2, 10, 13]. Clearly, if $U_0 > U^*$ and $r > 0$ then the solution approaches the equilibrium from above, i.e., the curve is falling, and if $U_0 < U^*$ and $r > 0$ then the curve is growing. If $r < 0$ then the solution approaches $U = 0$ (Fig 1, right).

The considerations about transferring epidemiological concepts above suggest that standard results from mathematical epidemiology can be transferred to the linguistic domain as well [41]. More specifically, this means the following: First, the basic reproductive ratio of linguistic constituents can be estimated given estimates of $\beta, \gamma$ and $\mu$, or intrinsic growth rate $r$ and any two of the three model parameters. Mortality rate $\mu$ can be estimated from demographic data (as $\mu = 1/LE$). Adoption rate $\beta$ and abandoning rate $\gamma$, however, are much harder to determine.

Second, for linguistic constituents at their population dynamic equilibrium, the basic reproductive ratio can be estimated by means of prevalence, i.e., the fraction of users. That is, $R_0 = 1/(1-p_U)$, where $p_U = U^*$ is prevalence at population dynamic equilibrium. Third, for linguistic constituents at their population dynamic equilibrium, the basic reproductive ratio can be estimated by means of age of acquisition, i.e., the age at which the constituent was learned, so that $R_0 = LE/AoA$, where $AoA$ is the age of acquisition of the constituent.

Fourth, we can use intrinsic growth rate to estimate the basic reproductive ratio as $R_0 = 1 + r \cdot L$ where $L = LE - AoA$ approximates the period during which an individual uses a constituent. This approximation only holds well if the constituent repertoire of individuals does not shrink substantially as they age.

Note that, theoretically, all ways of estimating the basic reproductive ratio should yield identical estimates of $R_0$. In particular, this should hold true for estimates based on age of acquisition and prevalence, since both share the same assumption, namely that the population

Table 2. **Epidemiological and linguistic variables relevant to the estimation of $R_0$.**

| Variable | Description |
|---|---|
| $AoI$ | Age of first infection of a disease (*epidemiological*) |
| $AoA$ | Age of acquisition of a constituent (*linguistic*) |
| $p_I = I^*$ | Equilibrium prevalence, i.e., fraction of infected individuals at endemic equilibrium (*epidemiological*) |
| $p_U = U^*$ | Equilibrium prevalence, i.e., fraction of users at population dynamic equilibrium (*linguistic*) |
| $r$ | Intrinsic growth rate of a disease (*epidemiological*) or constituent (*linguistic*) |
| $R_0$ | Basic reproductive ratio: <br> Expected number of individuals infected by an infected individual introduced into a population of susceptible individuals (*epidemiological*). <br> Expected number of non-users that adopt a constituent from a single user of that individual in a population of non-users (*linguistic*). |

dynamics rest at their equilibrium. Table 2 juxtaposes epidemiological and linguistic variables relevant for estimating the basic reproductive ratio.

Like the herd immunity threshold, we can derive, once $R_0$ is estimated, the stability threshold $H = 1 - 1/R_0$. In linguistic terms it denotes the minimum fraction of individuals which could potentially acquire a linguistic constituent that need to be excluded from adoption for that constituent to vanish in the long run. If there were, for instance, hypothetical political measures for banning a certain word, one would need to ensure that at least $H$ individuals are not able to adopt that word. Conversely, if, in this hypothetical scenario, $1 - H$ individuals would refuse to adhere to these measures, the spread of the word could not be prevented. Note that this does not consider social structure in any way. We will come back to this point in the discussion section.

## 2. Materials and methods

All computations described in this section were done with R (4.4.1). Data and code for this study are available at https://gitlab.com/andreas.baumann/basic_reproductive_ratio.

### 2.1 Data

We test the predictions outlined above on the lexical level, i.e., by inspecting dynamics of words. We restrict our diachronic analysis to words that have increased in frequency during the past two centuries, and we will refer to such words as *lexical innovations*. We do so irrespective of whether these words have already been present initially at low frequency, whether they are associated with new concepts or are involved in onomasiological competition, whether they are new word formations or have been adopted from a different language. Moreover, we only consider *periphery* words, which we define, somewhat simplistically, as words that are not used by everyone, based on prevalence estimates.

For obtaining estimates of the basic reproductive ratio for these words, we combine different data sets in different languages: English (eng), Spanish (spa), German (ger), and Italian (ita). We use AoA estimates for a large set of English from Kuperman et al. [42]. AoA norms for about 4600 Spanish verbs were taken from Alonso et al. [43]. For German, we used data from Birchenough et al. [44]; 3,200 words), and for Italian, data from Montefinese et al. [45] was employed. All of these estimates refer to word forms (there are sometimes separate AoA norms for inflectional and/or derivational forms of a base form such as *accelerate*, *accelerated*, *accelelation*, etc. in [42]) and they were obtained by averaging subjective age-of-acquisition ratings collected via (crowdsourcing) surveys. Although ratings are subjective, estimates

obtained by means of this methodology were shown to correlate with age-of-acquisition estimates obtained under controlled conditions [42, 46].

Lexical prevalence was proposed as a relevant covariate for psycholinguistic research [47], so that prevalence estimates are already available for a set of languages. Prevalence estimates for English word forms were taken from Brysbaert et al. (2019). These estimates were aggregated via crowdsourcing as well. Note that prevalence is not probit transformed as in Brysbaert et al. [48]. Rather, for a word, the prevalence estimate measures the fraction of individuals participating in the survey knowing the word. Baumann and Sekanina (2022) show that there is a slight difference between the fraction of individuals knowing a word (only passive knowledge) and the fraction of individuals using a word (active knowledge). The latter operationalization, as a matter of fact, fits better to prevalence as introduced in the previous section as the fraction of users. Overall, however, both estimates correlate strongly so that the estimates from Brysbaert et al. (2019) will serve as good enough proxies for the present study. For the other three languages (ger, spa, ita), we approximated prevalence by computing, for each word, the fraction of participants reporting a subjective AoA rating [43–45]. If no rating was given or if participants reported that the word was unknown this was interpreted as neither knowing nor using the word. Note that the accuracy of the prevalence estimates substantially depends on sample size, i.e., on the number of participants they are based on. The respective sample sizes are about 300 for English, about 23 for German, about 50 for Spanish, and 25 for Italian. Hence, maximal margins of error (for $p = 0.5$ at a 95% confidence level) range from about ±0.03 (for English) to slightly over ±0.1 (for German).

For measuring growth, we use word frequencies derived from diachronic corpus data. For English, we derive frequency trajectories from the Corpus of Historical American English (COHA; Davies 2010) for all words in the intersection of Kuperman et al. [42] and Brysbaert et al. (2019). This corpus spans the period from 1820 to 2000. Trajectories have a resolution of one frequency estimate per decade. Frequencies are normalized with respect to the decade-wise sub-corpus size. The main advantage of COHA clearly is that it is balanced with respect to genre. For German, Spanish, and Italian, we needed to resort to normalized frequency trajectories extracted from Google books (ngrams; https://books.google.com/ngrams/). For the sake of consistency, exactly those word forms represented in the respective AoA lists were chosen in all queries for both COHA and Google books. This is, because the AoA list in [42] features estimates for separate inflectional/derivational word forms and does not differentiate between word classes. Thus, there are, e.g., separate AoA estimates for *abbreviate* and *abbreviated*, but it is not clear whether the latter corresponds to the past-tense form of the verb or to the adjectival participle. Hence, we did not differentiate between word classes in the corpus data either and aggregated frequencies of all word forms matching the entries in the AoA list per decade in COHA. In Google books, frequency trajectories have a resolution of one estimate per year. Frequencies were aggregated per decade, however, so that the resolution matches that of COHA. Table 3 shows a summary of all data sources used in this study. Fig 2 displays prevalence estimates, age-of-acquisition estimates, and historical frequency trajectories of four lexical examples.

## 2.2 Methods

**2.2.1 Operationalization.** The formulae in Section 1.2 are used to estimate the basic reproductive ratio based on (i) age of acquisition, (ii) prevalence, and (iii) diachronic trajectories. We refer to (i-iii) as estimation methods. For (i-iii) we restrict ourselves to words with a prevalence smaller than 0.95, i.e., we exclude the core lexicon. In addition, for (iii), we focus on lexical innovations, which we conceptualize as words that have been increasing significantly

**Table 3. Data summary for all languages (eng, spa, ger, ita).**

| Variable | Language | Comment | Source |
|---|---|---|---|
| AoA & prevalence | eng | 30,000 words | [42, 48] |
| | ger | 3,200 words | [44] |
| | spa | 4,640 words | [43] |
| | ita | 1,957 words | [45] |
| Frequency (diachronic) | spa, ger, ita | Trajectories (years aggregated per decade) | Google ngrams accessed with ngramr [50] |
| | eng | Trajectories (decades) | COHA [49] |

in frequency throughout the observation period (see below for more details). We use the collected age-of-acquisition estimates, an estimate of life expectancy based on demographic data (https://data.worldbank.org/), and estimate the basic reproductive ratio as (i) $R_0^{AoA} = LE/AoA$. Note that life expectancy has increased through the past two centuries and that women have, on average, a higher life expectation than men, so that the assumption of a single point estimate is clearly simplistic. In the UK and the US, for instance, life expectancy has increased from about 71 in the early 1960s to about 80 in the 2000s (Fig 3, left). Notably, UK/US displays the smallest increase in life expectancy. For all populations, mean LE is roughly 75 with an average margin of error of 1 year (95% confidence interval) in this time period. This margin of error will be considered in the error analysis in 2.2.2. Furthermore, the estimation of the basic reproductive ratio depends on the age distribution being roughly rectangular (i.e., relatively evenly distributed; see above). Such a distribution is typically adopted in high-income countries (Fig 3, right) [31, 32]. In particular, this assumption seems to be met for the English-speaking population. More straight forwardly, we can use prevalence estimates $p_U$ to derive (ii) $R_0^p = 1/(1 - p_U)$.

Both of these estimates rest on the assumption that the population dynamics are at their equilibrium, i.e., that the fraction of users neither grows nor falls substantially. For that reason,

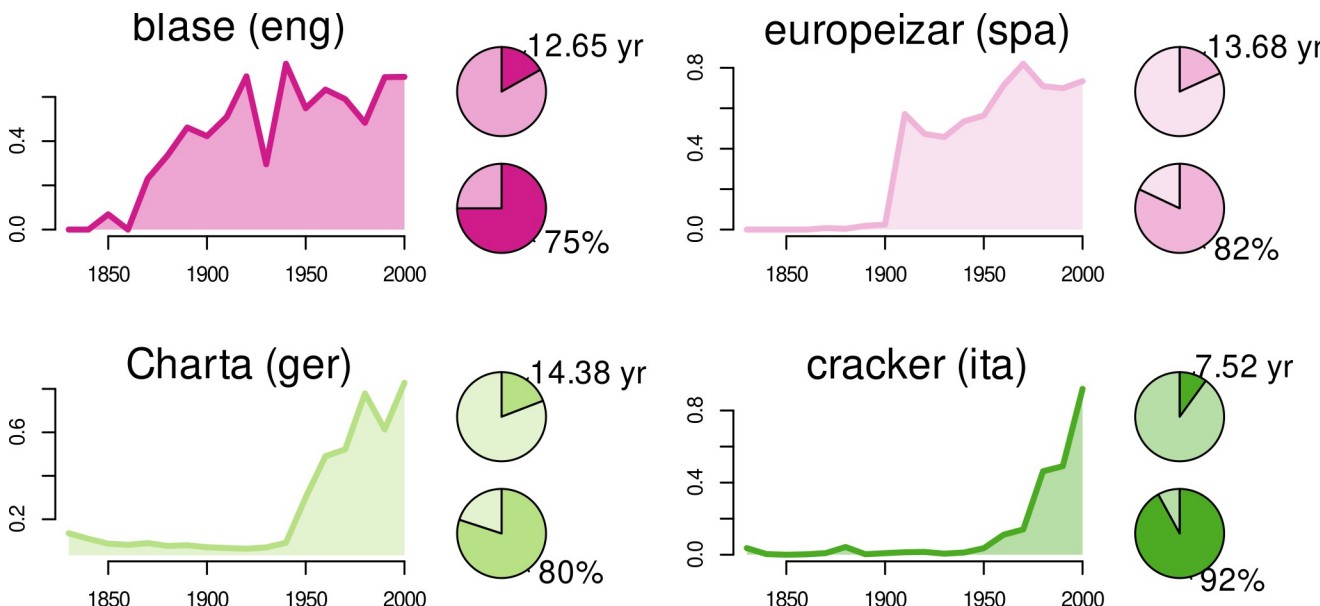

**Fig 2. Lexical innovations.** Examples for all considered languages: *blasé* (eng), *europeizar* (spa, 'europeanize'), *Charta* (ger, 'charter'), and *cracker* (ita, 'cracker'): trajectories of normalized frequency (cf. 2.2.1) and pie charts illustrating age of acquisition relative to life expectation (top) and prevalence (bottom).

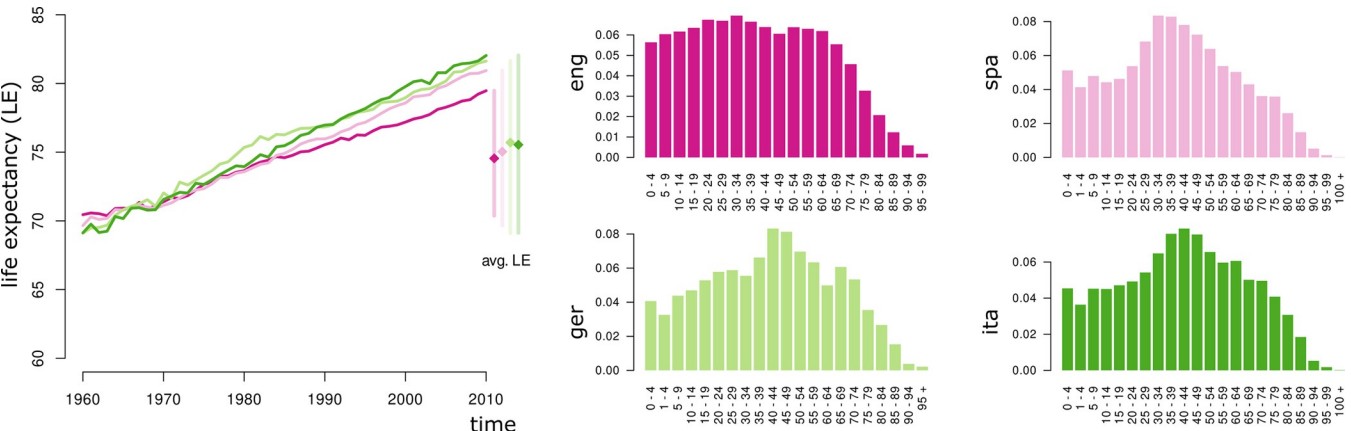

**Fig 3. Demographic information for the relevant speaker populations.** Left: life expectancy (LE) depending on time with mean LE (from left to right: eng, spa, ger, ita). Data taken from https://data.worldbank.org/. Right: age distributions for UK and US combined (eng), Spain (spa), Germany (ger), and Italy (ita) as representatives of the respective speaker populations. Data taken from https://data.un.org/.

we compute for each word the absolute value of the Spearman correlation coefficient of time and frequency, $\rho \geq 0$ from 1950 to 2000. We use $S = 1 - \rho$ as a measure of contemporary stability. If $S$ it is close to 1 for a word this indicates that the dynamics of that word are relatively stable and rest at their dynamic equilibrium. We will use this to examine the impact of $S$ on the quality of the estimates of the basic reproductive ratio.

Using frequency data to estimate the basic reproductive ratio requires a bit more work. The goal is to use the approximation $R_0 \cong 1 + r \cdot L$ where $r$ and $L$ denote intrinsic growth rate and the duration of the prolific period, respectively. As a matter of fact, to get $r$ we need a prevalence trajectory, i.e., a trajectory of the number of users $U(t)$ over time $t \in \mathcal{T}$, as opposed to a frequency trajectory. Here, $\mathcal{T} = \{t_0, t_1, \dots, t_{18}\}$ is the observation period divided into 18 decades (from 1820 to 2000). It is not at all easy to obtain word prevalence based on corpus data. This is because corpora often feature only a small number of authors and because the amount of text per author is likely not representative of their linguistic knowledge (let alone the fact that the author population in linguistic corpora is not necessarily representative of the whole population, in particular, as far as the social dimension is concerned). See Johns et al. [51] for an exploration of ways to estimate prevalence based on corpus data.

In the present study, however, we adopt a different approach to reconstruct prevalence trajectories for a subset of words. For each word, we take its frequency trajectory $f(t)$ and we restrict our set of words to those that have a present-day prevalence $p_U < 0.95$. For each word we check if it has increased significantly based on a Spearman's correlation test (correlating $U(t)$ and $t$ at $\alpha = 0.05$). Furthermore, we derive the stability coefficient $S$ as above. Subsequently, we normalize each frequency trajectory with respect to its maximum, i.e., $\tilde{f}(t) = f(t)/\max f(t)$, and we define a word's prevalence trajectory as $U(t) = p_U \cdot \tilde{f}(t)$. That is, we do as if the frequency trajectory would mirror the trajectory of prevalence. We only considered words with $U(t) \leq 1$ for all $t$ and significantly increasing prevalence trajectories (i.e., words that we consider lexical innovations in this study).

In a final step, we fit a logistic model $\hat{U}(t) = a/(1 - b \cdot e^{-rt})$, $a, b > 0$, as defined in Section 1.2 to the trajectory $U(t)$ in order to estimate intrinsic growth rate $r$ and define (iii) $R_0^r = 1 + r \cdot L$, where we take $L = LE - AoA$ as the prolific period of the word. This estimate was not computed for words with $1 + r \cdot L < 0$, since the basic reproductive ratio is conceptually

non-negative. For each model, we compute

$$R^2 = 1 - \frac{\sum_{t \in \mathcal{T}} (U(t) - \hat{U}(t))^2}{\sum_{t \in \mathcal{T}} (U(t) - \overline{U(t)})^2} \tag{5}$$

to assess goodness of fit, where $\overline{U(t)}$ denotes mean prevalence across all $t \in \mathcal{T}$. Finally, the inflection point $t_I$ is extracted from each model. This point divides the trajectory into an initial (accelerating) and final (decelerating) period and helps to identify when the spreading phenomenon took place, at least approximately.

Of course, this approximation is not unproblematic. First, it is likely that the prevalence trajectory of a word in fact precedes that of frequency because a word can also spread, language internally, through various possible linguistic contexts. That is, $r$ and hence $R_0^r$ is potentially underestimated (because $r$ should be greater than frequency developments suggest). On another account, life expectancy has changed considerably over the past two centuries so that $LE$ and hence $R_0^r$ is in fact lower than assumed. Perhaps these biases balance each other, but this still needs to be tested. Finally, age of acquisition is likely not constant over time. See Section 4.1 for discussion. Table 4 lists the sample sizes used for all ways of measuring the basic reproductive ratio in the four languages, considering all restrictions mentioned above (periphery words; lexical innovations).

**2.2.2 Statistical analysis.** We inspect the quality of the obtained estimates for the basic reproductive ratio, and hence the quality of the estimation methods, in three ways. First, we investigate, for each language, to what extent estimates gained via (i) age of acquisition, (ii) prevalence, and (iii) growth fall into a similar range. Second, we assess to what extent estimates provided by methods (i-iii) correlate. Finally, a derivation and analysis of the margin of error is provided for all estimates of the basic reproductive ratio.

In the first part, we compute medians of $R_0$ estimates for all methods (i-iii) in all languages, and assess if they differ from each other (Wilcoxon signed-rank test with $ES = Z/\sqrt{N}$ as effect size, where $N$ is sample size [52, 53], as well as their pairwise correlations (note that correlations with $R_0^r$ cannot be computed for Italian due to the lack of words with the required diachronic profiles). Second, we model pairwise associations between $R_0^r$, $R_0^p$, and $R_0^{AoA}$, controlling for stability $S$. For each language, we fit a Gaussian model (i+ii) $R_0^{AoA} = b_0 + b_p R_0^p + b_{pS} R_0^p S + \epsilon$, and for all languages except for Italian, we fit two additional Gaussian models, (ii+iii) $R_0^r = b_0 + b_p R_0^p + b_{pS} R_0^p S + \epsilon$ and (i+iii) $R_0^r = b_0 + b_{AoA} R_0^{AoA} + b_{AoAS} R_0^{AoA} S + \epsilon$. In the latter two models, data points are weighted by goodness of fit ($R^2$) of the temporal trajectory the respective growth rates were estimated from. This is to demote the impact of unreliable $R_0^r$ estimates when assessing the relationship with the other two measures, $R_0^p$ and $R_0^{AoA}$. Stability is controlled for because estimates of $R_0^p$ and $R_0^{AoA}$ rely on the assumption of the respective population dynamics resting at their equilibrium. Hence, correlations among measures of $R_0$ are expected to be higher when this assumption is met, i.e., when stability $S$ is high (close to 1). Italian was excluded in models (i+ii) and (ii+iii)

**Table 4. Sample sizes for all languages (eng, spa, ger, ita) and all estimation methods (i-iii).**

| Language | $R_0^{AoA}$ | $R_0^p$ | $R_0^r$ | total |
|---|---|---|---|---|
| English | 12494 | 12512 | 1774 | 12512 |
| German | 78 | 78 | 19 | 78 |
| Spanish | 1361 | 1361 | 330 | 1361 |
| Italian | 32 | 32 | 1 | 32 |

involving $R_0^r$ as outcome variable, because there was only a single valid estimate derived from diachronic frequency data (see above).

The formulas used for estimating the basic reproductive ratio, (i) $R_0^{AoA} = LE/AoA$, (ii) $R_0^p = 1/(1-p)$, and (iii) $R_0^r = 1 + r \cdot L = 1 + r \cdot (LE - AoA)$, involve nonlinearities and multiple parameter estimates that need to be considered when providing an estimate of the margin of error of $R_0^{AoA}$, $R_0^p$, and $R_0^r$. This is important since life expectancy, age of acquisition, and prevalence represent estimates themselves. We employ error propagation through the first-order Taylor expansion of $R_0^{AoA}$, $R_0^p$, and $R_0^r$ to obtain the respective margins of error, $\Delta R_0^{AoA}$, $\Delta R_0^p$, and $\Delta R_0^r$, i.e., one half of the corresponding 95% confidence interval [54, 55]. More specifically, we have

$$\Delta R_0^{AoA} = \sqrt{\left(\frac{\partial R_0^{AoA}}{\partial LE} \cdot \Delta LE\right)^2 + \left(\frac{\partial R_0^{AoA}}{\partial AoA} \cdot \Delta AoA\right)^2} = \sqrt{\left(\frac{\Delta LE}{AoA} \cdot\right)^2 + \left(\frac{LE \cdot \Delta AoA}{AoA^2}\right)^2}, \quad (6)$$

$$\Delta R_0^p = \sqrt{\left(\frac{\partial R_0^p}{\partial p} \cdot \Delta p\right)^2} = \frac{\Delta p}{(1-p)^2} \quad (7)$$

and

$$\Delta R_0^r = \sqrt{\left(\frac{\partial R_0^r}{\partial r} \cdot \Delta r\right)^2 + \left(\frac{\partial R_0^r}{\partial LE} \cdot \Delta LE\right)^2 + \left(\frac{\partial R_0^r}{\partial AoA} \cdot \Delta AoA\right)^2}$$

$$= \sqrt{\left((LE - AoA)\Delta r\right)^2 + r^2(\Delta LE^2 + \Delta AoA^2)} \quad (8)$$

where $\Delta LE, \Delta AoA, \Delta p$, and $\Delta r$ are the respective margins of error of life expectancy, age of acquisition, prevalence, and growth rate. For life expectancy, we assume a margin of error of 1 (see 2.2.1). For age of acquisition, the standard error is computed as $SE = \sqrt{\sigma/N_{AoA}}$, where for each word standard deviation $\sigma$ and sample size $N_{AoA}$ are directly taken from the four age-of-acquisition datasets. For prevalence, the standard error is computed as $SE = \sqrt{p(1-p)/N_p}$. Here, sample size $N_p$ is again inferred from the employed datasets. Note that in the case of English, $N_{AoA}$ and $N_p$ differ from each other since measures were taken from two different resources [42, 48]. For age of acquisition and prevalence, the margin of error is then computed as 1.96·$SE$.

Obtaining the margin of error $\Delta r$ requires a little more work since growth rate $r$ is estimated from a logistic model fitted to a trajectory and not derived from a closed formula. Moreover, the trajectory $U(t) = p \cdot \tilde{f}(t)$ itself depends on estimates of (contemporary) prevalence weighted by occurrence frequency in diachronic corpora, as described above. We opted for bootstrapping the margin of error for each word-specific growth rate $r$ in the following way. For each time step $t$, we sampled $n = 300$ proportions from a binomial distribution $B(N_p, U(t))$. This resulted in $n$ trajectories $\tilde{U}(t)$ randomly sampled around $U(t)$. Each of these trajectories was in turn used to fit a logistic growth model as described above in 2.2.1, from which a logistic growth rate $\tilde{r}$ was derived. Subsequently, the distribution of all logistic growth rates $\tilde{r}$ obtained in this manner was used to compute the bootstrapped 95% confidence interval of the estimate $r$. The margin of error $\Delta r$ was finally taken as one half of the width of this confidence interval. This was done for each word for which growth rate was estimated in 2.2.1. Note that this procedure still ignores the fact that occurrence frequencies used for constructing trajectories represent point estimates as well. Put differently, we need to assume (a) that all employed text

corpora are so large that the corresponding errors vanish (which is plausible), and (b) that the employed text corpora are representative samples of the examined languages (which is arguably debatable).

The obtained margins of error were then used together with Eqs ([6–8]) to compute the margins of error $\Delta R_0^{AoA}$, $\Delta R_0^p$, and $\Delta R_0^r$ for each word. Finally, the distribution of inflection points $t_I$ was examined descriptively for each language in order to assess when the observed lexical developments took place.

## 3. Results

The analysis of expected values for the basic reproductive ratio (for all languages and methods) shows that estimates typically range from about 3 to about 14 ([Fig 4]; inter-quartile ranges go from at least about 2 (ger: $R_0^r$) to at most about 17 (ger: $R_0^p$)). Medians are shown in [Table 5]. There are no clear biases visible as far as the methods are concerned. Using prevalence yields higher estimates for the basic reproductive ratio than using age of acquisition does. This is the case for English (small effect) and Italian/German (large effects), but not for Spanish. Using growth rates to estimate the basic reproductive ratio yields smaller estimates for three languages (Spanish, German, Italian), but not for English. Moreover, the distributions of estimates based on growth for English and Spanish display a long right tail with $R_0^r$ going well above 20 for a set of words. Most estimates fall below about 14 in both languages, however. German prevalence-based estimates are higher than this is the case for all other languages, and

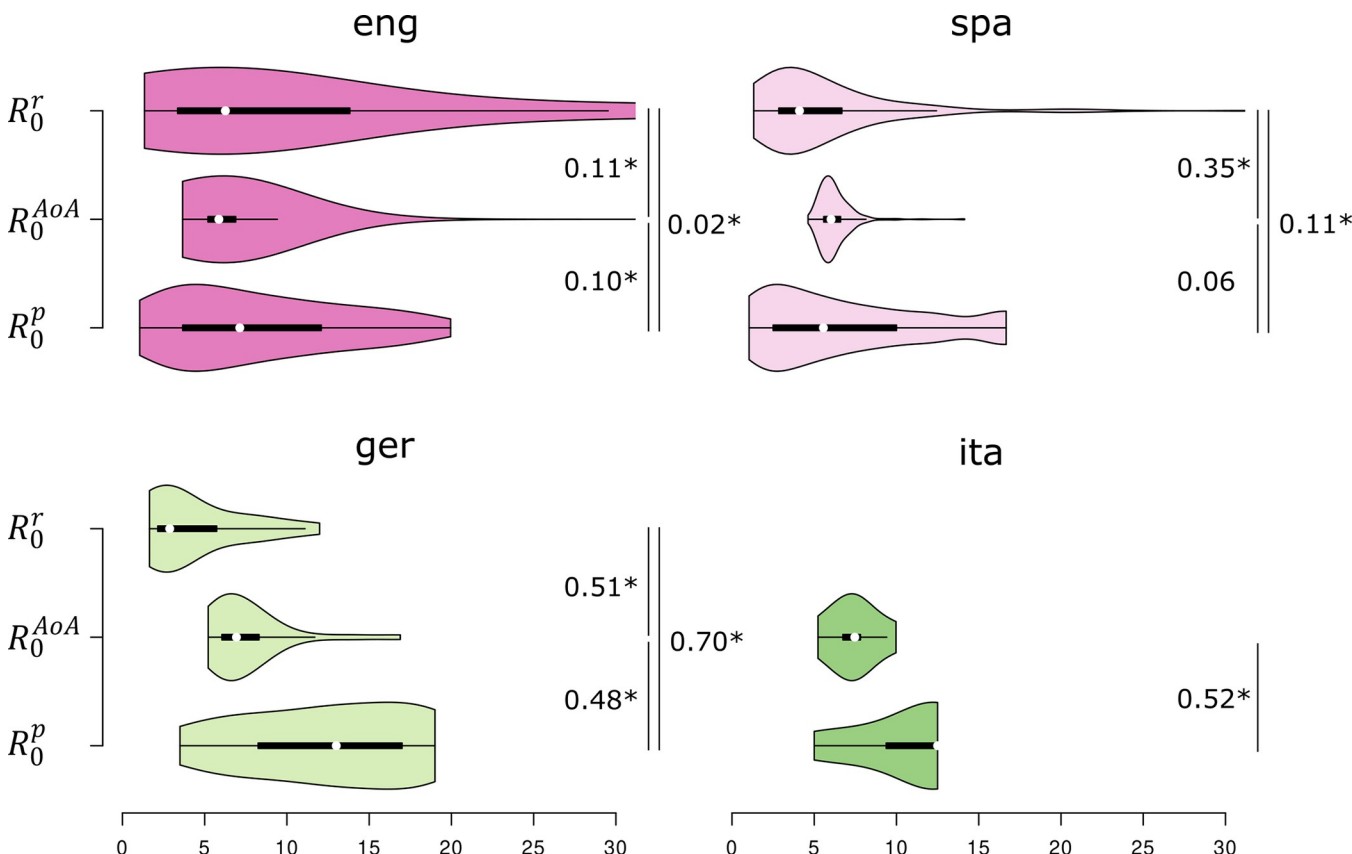

**Fig 4. Distributions of all $R_0$ estimates in all languages.** White circle: median; black bar: inter-quartile range. Pairwise differences (normalized effect size) are displayed to the left of each violin chart ('*' indicates significantly non-zero differences at $\alpha = 0.05$).

**Table 5. Median estimates for all methods (i-iii) of estimating the basic reproductive ratio in all languages.**

| Language | median $R_0^{AoA}$ | median $R_0^p$ | median $R_0^r$ |
|---|---|---|---|
| English | 5.7 | 5.9 | 6.2 |
| German | 7.2 | 14.0 | 2.6 |
| Spanish | 6.0 | 5.0 | 3.2 |
| Italian | 7.0 | 12.5 | 5.4 |

the assessment of the range of $R_0^r$ in Italian is obviously imprecise given that there is only a single point estimate. Is worth to point out that $R_0^{AoA}$ estimates seem to be relatively consistent in the sense that average estimates for all four languages fall into a small range (from 5.7 to 7.2). Of all four languages, English produces the most consistent estimates ranging between 5.7 and 6.2 with only small differences in terms of normalized effect size. Overall, there seems to be a positive relationship between consistency and sample size, English and Spanish yielding more consistent estimates across the three methods than the other two languages with considerably smaller sample sizes.

Fig 5 shows pairwise correlations among the three measures for English, Spanish, and German, but without taking differences in stability into account. Relationships among $R_0^p$ and $R_0^{AoA}$ are positive in all three cases (mildly positive correlation coefficients between 0.19 and 0.30).

The results of the models accounting for pairwise relationships among $R_0^p$ and $R_0^{AoA}$ including an interaction with stability are shown in Fig 6. The relationships are positive for English, Spanish, and German, which is what one would expect, given the theoretical considerations. In the case of Italian, no clear relationship can be found. Interestingly, the association between $R_0^p$ and $R_0^{AoA}$ is not substantially modulated by the stability measure $S$. Words that are learned early are also those that are used by a large fraction of individuals, irrespective of whether they have been used stably in the past decades. However, at least for English and Spanish, a

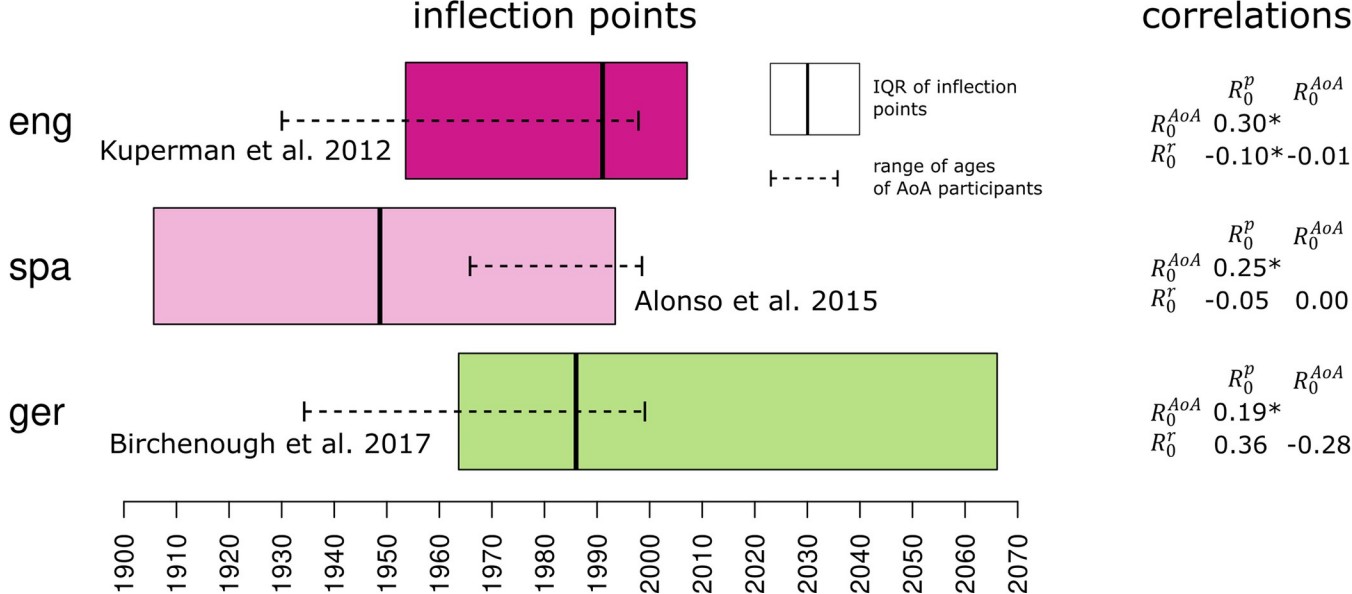

**Fig 5. Distributions of inflection points and correlations among all $R_0$ estimates in three languages (eng, spa, ger).** Left: boxes show interquartile ranges of inflection points $t_I$ per language. Dashed lines represent ranges of participant age (displayed as birth dates) in each of the three age-of-acquisition surveys (see Table 3). Right: Pearson correlation coefficients among all $R_0$ measures per language ('*' indicates significantly non-zero differences at $\alpha = 0.05$).

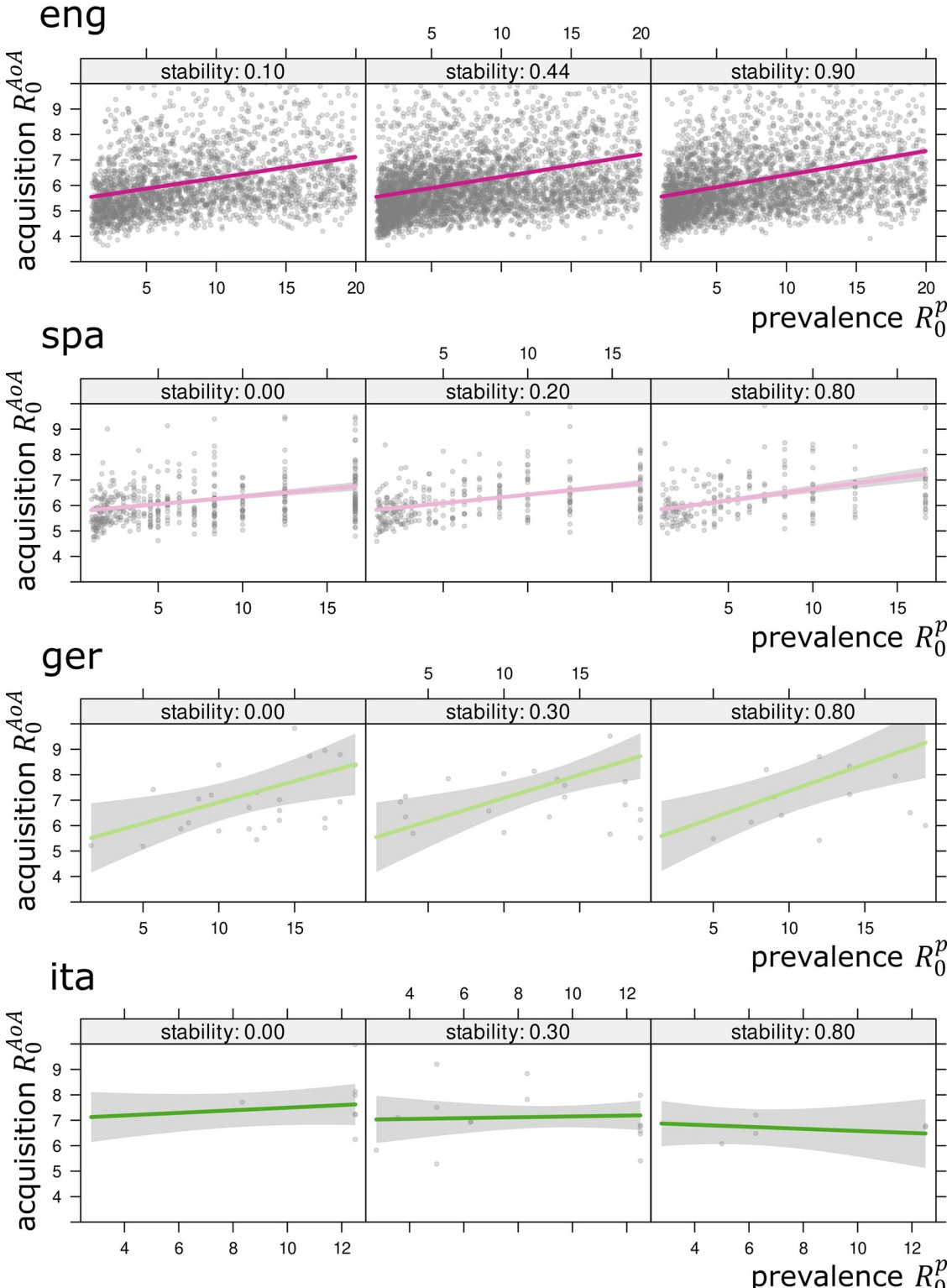

**Fig 6. Models (i+ii).** Linear models (i+ii) of $R_0^{AoA}$ depending on $R_0^p$ controlled by stability $S$ with 95% confidence regions. For all languages but Italian there are statistically robust positive associations between $R_0^{AoA}$ and $R_0^p$. In each model visualization, the region close to $S = 1$ corresponds to those words that are contemporarily stable ($R_0^{AoA}$ and $R_0^p$ more reliable).

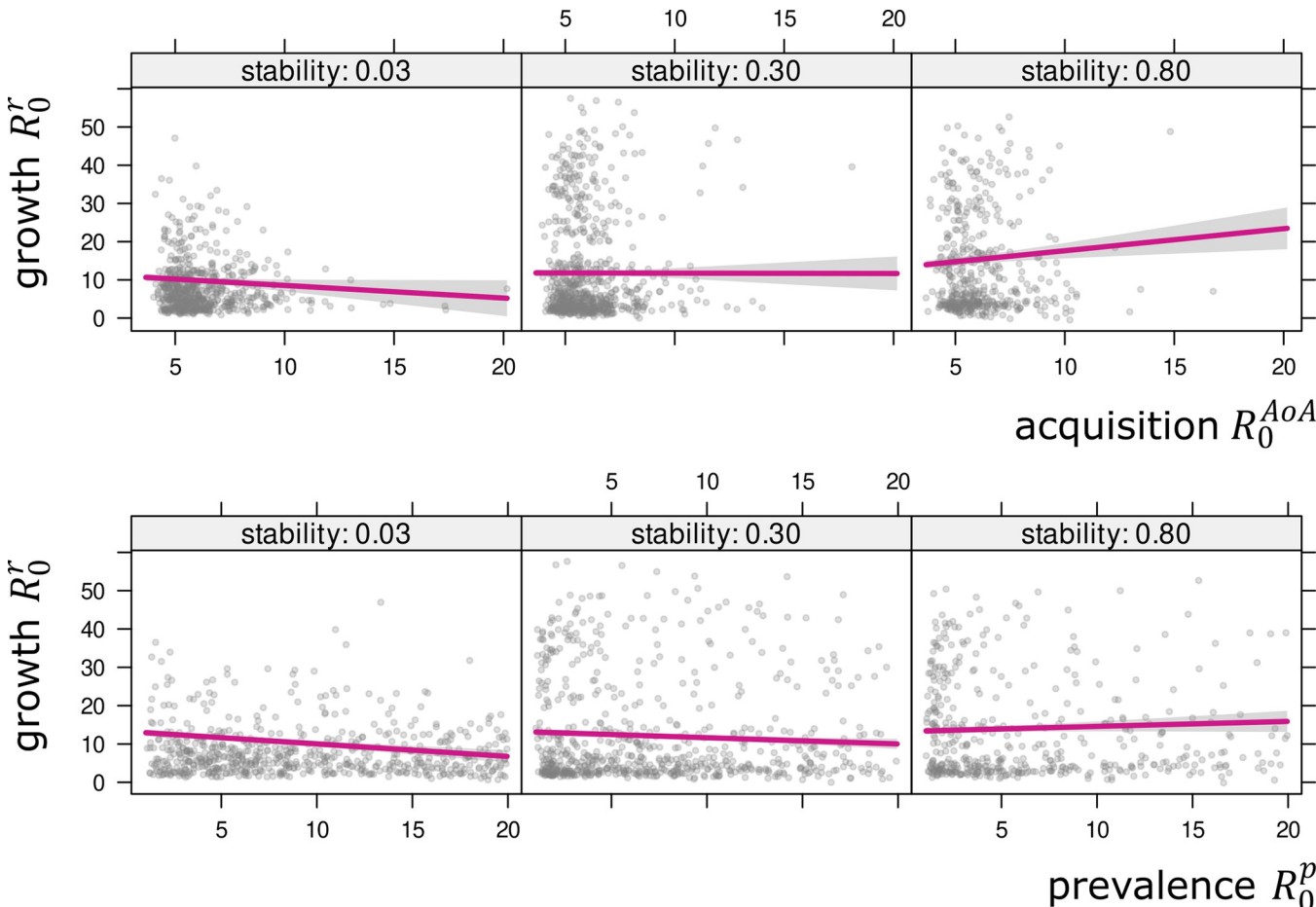

**Fig 7. Models (i+iii) and (ii+iii) for English.** Linear models (i+iii) of $R_0^r$ depending on $R_0^{AoA}$ (top panel) and (ii+iii) of $R_0^r$ depending on $R_0^p$ (bottom panel), controlling for stability $S$, with 95% confidence regions. Each visualization is split into three panels, reflecting increasing stability (from left to right). In the case of English, there are significantly non-trivial interactions: positive effects for high $S$, and negative effects for low $S$, as expected.

significantly positive interaction coefficient $b_{pS}$ can be determined. That is, the positive relationship between $R_0^p$ and $R_0^{AoA}$ is slightly strengthened as stability $S$ increases. So, stability seems to have a small but positive effect on the quality of the estimates. This is what one would expect given the assumptions of the formulas for deriving $R_0^p$ and $R_0^{AoA}$. In both cases, $S$ needs to be sufficiently high. In other words, a lacking positive relationship between ways of estimating the basic reproductive ratio for low-stability words, e.g., for words that contemporarily show an increasing trajectory, is in fact not at all surprising. The models assessing the relationship between $R_0^r$ and $R_0^{AoA}$ and $R_0^r$ and $R_0^p$ reveal a different picture (Fig 7). In fact, weakly positive associations can only be found for English, and only for high-stability words. For words that are either in the process of growing or declining (low $S$), the relationship is even reversed. For the other languages, no statistically robust effects can be found. All model coefficients are listed in Tables 6–8.

Fig 8 shows the distribution of margins of error across all languages and estimation methods. In the case of English and Spanish, margins are relatively small (with median scores not exceeding 1.28 for English and 2.77 for Spanish), irrespective of how it was estimated. For example, the confidence interval for the basic reproductive ratio of an average English word in the dataset goes from about 4 to about 8. The German and Italian data, however, reveal large

**Table 6. Models (i+ii) with $R_0^{AoA}$ as dependent variable.**

| Language | Coefficient | Estimate | SE | p-value |
|----------|-------------|----------|-----|---------|
| eng | $b_0$ | 5.459 | 0.028 | < 0.001* |
|  | $b_p$ | 0.081 | 0.004 | < 0.001* |
|  | $b_{pS}$ | 0.014 | 0.006 | 0.009* |
| spa | $b_0$ | 5.757 | 0.028 | < 0.001* |
|  | $b_p$ | 0.060 | 0.004 | < 0.001* |
|  | $b_{pS}$ | 0.037 | 0.014 | 0.004* |
| ger | $b_0$ | 5.256 | 0.764 | < 0.001* |
|  | $b_p$ | 0.166 | 0.061 | < 0.01* |
|  | $b_{pS}$ | 0.055 | 0.062 | 0.373 |
| ita | $b_0$ | 6.987 | 0.605 | < 0.001* |
|  | $b_p$ | 0.051 | 0.059 | 0.396 |
|  | $b_{pS}$ | -0.114 | 0.087 | 0.203 |

margins of error when the basic reproductive ratio is derived from prevalence. This is a direct consequence of the small sample sizes in the German (23 data points per prevalence estimate) and Italian data (25 data points per prevalence estimate). Across all languages, acquisition-based estimates are most accurate (median scores of the margin of error not exceeding 1.17).

## 4. Discussion

### 4.1 Generalizability and limitations

The observation that $R_0$ estimates derived through AoA and prevalence correlate for all languages (except Italian) is reassuring but not particularly surprising, for that matter. We will discuss the theoretical implications of this relationship below. What is surprising, however, is that the positive relationship between both estimates is not substantially modulated by the stability of the recent development. This is interesting because contemporarily unstable, i.e., growing or declining words, are not expected to yield reliable $R_0^p$ and $R_0^{AoA}$ estimates. Contemporarily unstable words do not fulfil a crucial assumption for deriving these measures (cf. 1.2 and 2.2.1). However, at least for English and German the interaction with stability points into the theoretically expected direction (albeit at a low effect size). The general lack of a positive correlation seems surprising given that $R_0^r$ was also defined by means of prevalence (because trajectories were normalized to fit present-day prevalence) and age of acquisition (because the prolific period is constrained by age of acquisition) in the first place.

**Table 7. Models (i+iii) with $R_0^r$ as dependent variable.**

| Language | Coefficient | Estimate | SE | p-value |
|----------|-------------|----------|-----|---------|
| eng | $b_0$ | 11.859 | 1.026 | < 0.001* |
|  | $b_{AoA}$ | -0.361 | 0.166 | 0.030* |
|  | $b_{AoAS}$ | 1.172 | 0.154 | < 0.001* |
| spa | $b_0$ | 7.255 | 1.813 | < 0.001* |
|  | $b_{AoA}$ | -0.271 | 0.303 | 0.373 |
|  | $b_{AoAS}$ | 0.243 | 0.162 | 0.136 |
| ger | $b_0$ | 10.521 | 4.957 | 0.050* |
|  | $b_{AoA}$ | -0.873 | 0.755 | 0.265 |
|  | $b_{AoAS}$ | -0.200 | 0.523 | 0.707 |

**Table 8. Models (ii+iii) with $R_0^r$ as dependent variable.**

| Language | Coefficient | Estimate | SE | p-value |
|---|---|---|---|---|
| eng | $b_0$ | 13.276 | 0.489 | < 0.001* |
| | $b_p$ | -0.340 | 0.056 | < 0.001* |
| | $b_{pS}$ | 0.589 | 0.111 | < 0.001* |
| spa | $b_0$ | 6.230 | 0.493 | < 0.001* |
| | $b_p$ | -0.059 | 0.057 | 0.301 |
| | $b_{pS}$ | 0.099 | 0.183 | 0.585 |
| ger | $b_0$ | 2.081 | 1.727 | 0.246 |
| | $b_p$ | 0.177 | 0.139 | 0.221 |
| | $b_{pS}$ | 0.393 | 0.308 | 0.220 |

The observation that most average $R_0$ estimates fall into a similar range, across languages as well as across estimation methods, is reassuring as well and potentially more revealing. While it can be expected that, across languages, scores derived with the same method fall into similar ranges, it is potentially more surprising that empirically estimated ranges indeed roughly coincide when making comparisons across methods. It can be seen that different estimation methods produce more similar results for English, followed by Spanish. They show certain discrepancies for German and Italian where prevalence-based estimates are higher than the other two estimates (large effect sizes in all cases for German and Italian). It is plausible that this can be attributed to the size and quality of the German and Italian data. Both datasets are relatively small, perhaps biased towards the core lexicon, and prevalence estimates were derived only indirectly (by considering missing AoA scores) and based on a smaller number of participants per word (about 25). Indeed, prevalence-based estimates for German and Italian are almost rendered useless by their, on average, large margins of error. We learn from this that using prevalence for the estimation of the basic reproductive ratio seems to only make

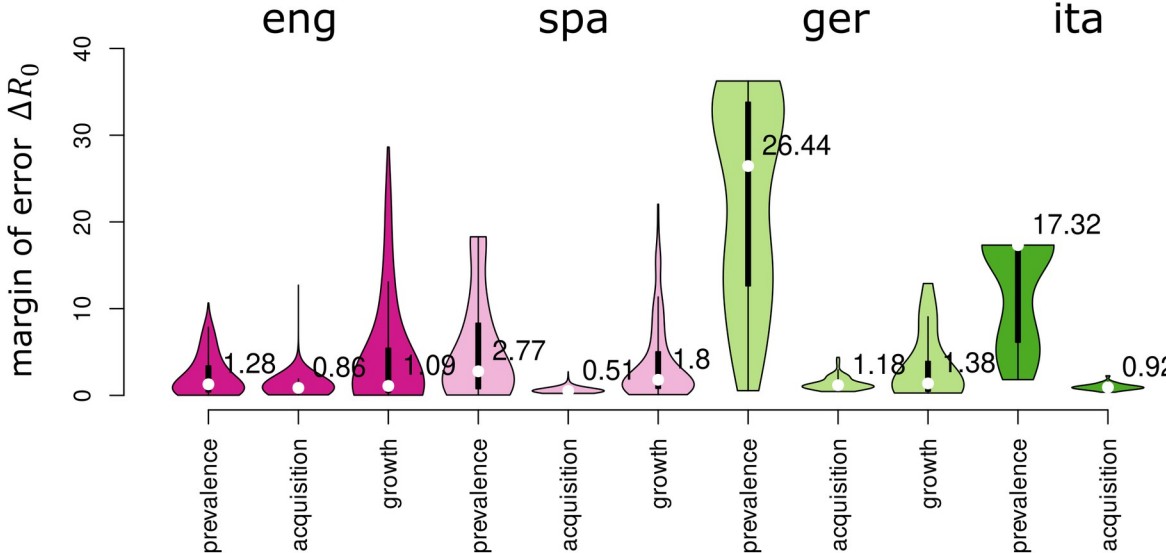

**Fig 8. Distributions of the margin of error of the basic reproductive ratio across all languages and estimation methods.** Margins of error based on a 95% confidence level. In the Italian data, only a single growth rate was derived, hence no distribution is shown for this configuration. Displayed numbers and white circles represent median scores. Thick black bars indicate inter-quartile ranges.

sense if prevalence estimates are based a sufficiently large number of participants (about at least 50 as in the case of Spanish).

The approach of estimating $R_0^r$ employed here suffers from a multitude of problems. First, it is very likely that age of acquisition, just like life expectancy, was not constant through the past two centuries. This is so, because age of acquisition is a function of frequency even if one controls for prevalence [48]. It could be that individuals living, say, one century ago were exposed much more rarely to words that are nowadays used frequently; not because these words were employed by fewer individuals but because they displayed low utterance frequency *per se* (for instance because there was a relatively more frequent competing synonym available). By consequence, historical age of acquisition would be higher, hence lowering $R_0^r$ for low-frequency words (because the prolific period is shortened in this way). That is, our approach potentially overestimates $R_0^r$ for low-frequency words.

The only way of mitigating this issue is to obtain reliable historical age-of-acquisition estimates. Evidently, subjective ratings are not available for early periods so that one needs to rely on reconstructed estimates, ideally working with evidence independent from frequency. An approach that might prove useful in this regard is that of estimating age of acquisition through simulating semantic learning. Botarleanu et al. [56] have exploited word embeddings trained on incrementally increasing corpora to approximate age of acquisition through semantic trajectories. Since this approach could be applied to historical corpora as well it could be used to approximate historical age of acquisition norms. Note, though, that word-embedding similarities were shown to depend on frequency as well [57] so that this factor needs to be considered carefully.

It is work pointing out, however, that judging from the range of inflection points of the estimated diachronic trajectories (Fig 5, left), the majority of all innovations covered in this study seem to represent relatively recent phenomena, mostly confined to the 20<sup>th</sup> century. The interquartile range of inflection points largely overlaps with the range of birth dates of the participants in the three age-of-acquisition surveys. In the case of English and German, participants seem to have witnessed the emergence of the examined innovations (in the case of Spanish, inflection points largely precede participant birth dates, though). Note that the actual inflection points could be overestimated due to the time lag between writing and publishing texts [58]. This affects Google Books (spa, ger) and, to a lesser extent, COHA (eng).

Second, and related to that, we used normalized frequency as a proxy of prevalence for reconstructing prevalence trajectories. While it was shown that lexical prevalence and frequency are correlated significantly [47, 59], the two measures do not match (for instance, computing Pearson's r for frequency and prevalence for all words in Brysbaert et al. [48] known by less than 95% of all individuals yields a correlation of $R = 0.28$. For prevalence thresholds of 80% and 50%, one yields $R = 0.16$ and $Rr = 0.12$, respectively). Notably, this relationship becomes weaker for lower prevalence values so that using frequency as a proxy does not seem to be a very reliable strategy when examining dynamics of lexical innovations that are initially rare by definition. More reliable historical prevalence estimates could be obtained by inspecting the fraction of authors in large-scale corpora that employ a particular word, an approach already pursued by Johns et al. [51] and Feltgen [60]. Although promising, this approach evidently suffers from data scarcity in earlier periods: for estimating prevalence in this way, one ideally needs both a large number of authors and a large and representative selection of texts per author, and prevalence will be necessarily underestimated [60].There is an interesting mismatch between the lacking correlation between $R_0^r$ and $R_0^{AoA}$ on the lexical level, and that on the phonological level that we have demonstrated to hold for the dynamics for English word-final consonant clusters [28]. The obvious advantage of phonological diachronic analyses is that

prevalence estimates are more accurate: before the onset of certain sound changes in the history of English, prevalence of (most) English word-final consonant clusters was exactly zero, and upon completion prevalence can be safely assumed to be close to one for the majority of such sound sequences. Estimating intrinsic growth then amounts to inspecting the steepness of the growth curve; a task involving fewer degrees of freedom.

Third, the nature of the employed historical corpus data could have affected the $R_0^r$ estimates. In contrast to COHA, which consists of a genre-stratified sample of texts throughout (most of) the observation period, the Google Books corpus was shown to be increasingly biased towards scientific texts [58]. As a result, growth rates $r$ and hence $R_0^r$ could be overestimated for scientific words and, consequently, underestimated for non-scientific words for Spanish and German (Italian was not included in the diachronic analysis, anyway). This also entails, that some non-scientific words were missed out in the present analysis due to not displaying significantly increasing trajectories. On another account, this bias might also explain why $R_0^r$ is significantly lower than $R_0^p$ and $R_0^{AoA}$ in the case of Spanish and German. A post-hoc comparison of English frequency trajectories in COHA and Google Books (see attached code for details) suggests that the great majority of trajectory pairs (78.8%) are relatively similar (strong and significant correlation between the pair of trajectories) while only a few cases (1.6%) yield opposing trends (significant negative correlation between the pair of trajectories). However, biases towards scientific vocabulary could still affect the estimates. With that in mind, conclusions drawn from English corpus data (i.e., COHA) seem to be more robust.

Fourth, the approximation of the prolific duration as $L = LE−AoA$ rests on the assumption that individuals do not forget words. Estimates of vocabulary size depending on age could make the approximation more robust (and, at the same time, inform us about the rate $\gamma$ in the model given in Eq (2)). Extant research on the relationship between age and vocabulary size, however, suggests that the number of known words increases at least up to the age of 70 [61, 62] so that the assumption made, although certainly simplifying, does have some empirical support.

The mathematical model is evidently simplistic. First, it assumes constant homogeneous mixing, and hence the three ways of estimating the basic reproductive ratio rest on this assumption as well. Problems that come along with this assumption are reviewed in [35]. It was shown that the invasion boundary in models that assume a more realistic population structure (scale-free networks) is lowered, so that $R_0$ estimates could be effectively underestimated [7, 63]. This is because in real-world social networks there are individuals that have a disproportionally large number of contacts [64, 65], an observation that was suggested to be responsible for why cultural evolution progresses faster than biological evolution [66]. Mean-field models provide a good way of integrating a more realistic network structure [7, 63]. Meyers [67] shows that in an epidemiological contact network,

$$R_0 = T\left(\frac{\langle k^2 \rangle - \langle k \rangle}{\langle k \rangle}\right), \tag{9}$$

where $T$ is transmissibility (i.e., mean probability of transmission given a contact event), and $\langle k \rangle$ and $\langle k^2 \rangle$ are mean degree and mean squared degree of the contact network. Since degree variance is given by $\langle k^2 \rangle - \langle k \rangle^2$, more heterogeneous contact networks are expected to display larger $R_0$ values. Network models could be informed by social-media data [68] or by extrapolating from contact surveys conducted for epidemiological purposes [69]. It would be interesting to see if a more realistic population structure would also lead to stronger correlations among the three $R_0$ estimates.

The model does not feature age structure. That is, adoption is assumed to be independent from age, which does not reflect how cognitive-semantic plasticity changes over time [70, 71]; clearly, the lexicon grows as age increases which in turn affects learning of new words. Likewise lexical processing and retrieval changes with age [72, 73]. Implementing several age-classes with differential adoption, abandoning, as well as mortality rates could mitigate this issue. Related to that, the procedure of estimating the basic reproductive ratio from age of acquisition assumes a rectangular age distribution. While this approximately applies to the age distributions of the present speaker populations considered in this study, this might not be the case for other speaker populations. On that note, the fact that English data mostly yield expected outcomes in this study (in contrast to the other three languages) might not necessarily be a reflex of larger sample size. It could also result from the fact that the age-distribution of the English-speaking population comes closer to a rectangular (i.e., equal) distribution than that of the other three languages (Fig 3). Likewise, life expectancy for the English-speaking population shows the smallest increase of all languages covered in this study. Both is in favor of the model assumptions. Hence, it could be that the higher quality of $R_0$ estimates (as is evident from the smaller differences between estimation methods and higher correlations among the respective estimated values) is grounded in demographic properties of the speaker population as well.

The model does not feature social structure, nor does it reflect geo-political patterns or other population demographics. Evidently, this can be problematic given that the investigated languages are spoken in several countries. However, compartment models are flexible enough to integrate information like this if contact rates among social and geo-political compartments are known. Likewise, the simplistic assumption of constant population size can be relaxed (note that work by May and Anderson [32] implies that similar approximations of the basic reproductive ratio based on age of acquisition can be used for increasing populations). Finally, the model assumes constant demographic and transmission rates. In particular, it excludes any type of stochasticity [16, 74]. Based on analytical work on stochastic SIS-models [75], we have shown earlier [76] that stochasticity in transmission decreases the basic reproductive ratio. This could be examined empirically based on diachronic corpus data.

Finally, when it comes to language-internal dynamics, the model is simplistic in that it assumes the transmission of one word to be independent from the rest of the lexicon. That is, it ignores effects of the semantic neighborhood [77] and semantic competition [78]. Such interactions would require a version of the model in which multiple words can explicitly interact with each other.

All the limitations discussed in this section point at avenues for improvement of the model and empirical analysis. Note, though, that similarly abstract models and concepts (in particular, the basic reproductive ratio) have been employed fruitfully in the field of epidemiology. This is by no means to be interpreted as an excuse for further improvement, but it tells us that we can learn something about contact-driven dynamics even from abstract models.

## 4.2 Implications

Given the limitations discussed in the previous section, does the approach adopted here represent an accurate model of lexical spread? Certainly not–as it depends on simplistic assumptions. If the model assumptions are met, however, our analysis suggests that periphery words, on average, show a basic reproductive ratio of about $R_0 \cong 6$, with inter-quartile ranges going from about 2 to 17, depending on the language and estimation method. This puts spreading periphery words–in principle–roughly on par with infectious diseases like Rubella ('three day measles', [79]) Poliomyelitis [24, 32], or the COVID-19 Delta variant [80, 81] ('in principle' because the basic reproductive ratio is dimensionless; see discussion below). This figure

defines a threshold that an average lexical innovation needs to pass to spread successfully [40]. That not every linguistic variant will spread is plausible, and the issue that new variants require a certain frequency of users in order to be of communicative value (which is rarely the case for new variants to begin with) was dubbed the "threshold problem" by Nettle [82]. Importantly, though, the epidemiological model adopted here does not feature any mechanism of positive frequency dependence modeling increased communicative value for higher user frequency. Rather, the model entails that a threshold for successful spread exists even if such frequency-dependent pressures are absent. Note that while the three estimation methods are expected to yield similar estimates, differences between languages might in fact reflect differential social, geographic, and demographic structures of the respective speaker populations. A larger set of languages is required, though, to investigate such effects.

The estimate also implies that the herd-immunity threshold of an average lexical innovation equals about 83% ($H = 1 - 1/R_0 = 1 - 1/6 = 0.83$; minimal and maximal bounds of observed inter-quartile ranges yield estimates for this threshold that go from 50% to 94%). That is, 83% of all individuals would need to be prevented from adopting that word for it to vanish in the long run. Whichever hypothetical measures of language policy a ruling body would implement, they would need to be fierce to show an effect. Likewise, only a fraction of 17% resisting said measures would suffice to maintain that word, even in the lack of social or geographic structure that would enforce stable usage in a smaller subpopulation. The observation has interesting consequences for the study of language policy and censorship ([83]; see also [84] for a recent example). What the model entails is that, from this admittedly simplistic epidemiological point of view, top-down prohibitive measures of language policy are likely not to have good chances of success (and note that in this argument we do not even consider the preserving effect of written language). Integrating social structure into the model (and potentially differential language policies) would be crucial for a detailed analysis of the matter.

Finally, the estimate $R_0 \cong 6$ also tells us about the role that individuals play in the propagation of new words through contact networks. Mean personal network size was shown to equal about 600 [85]. This reflects all individuals that one gets to know during one's life time. This means that a single individual that picks up a lexical innovation passes it on to only about 1% of their personal contacts–notably, throughout the whole period during which that individual knows and uses that word. If one considers an average offspring count of about 1 (per individual), the impact that a single individual has on the spread of words decreases even further: literally only a handful of other people will on average pick up the lexical innovation due to interactions with that individual. That is, the basic reproductive ratio of words can shed light on how information is transmitted through speaker networks [86, 87].

A notable difference between linguistics and epidemiology in this context is that the prolific, i.e., infectious, period of words $L$ is much longer than that of the above-mentioned infectious diseases (once acquired, words are kept for a much longer time span than, say, COVID-19 pathogens that get dealt with by the immune system over the course of a few weeks, on average). It follows from the approximation $R_0 \cong 1 + r \cdot L$ that words (with lower $r$ but higher $L$) can plausibly display similar $R_0$ values as the mentioned diseases (with higher $r$ but lower $L$). Low growth rates do not entail that lexical change *per se* is a rare phenomenon (we look at each individual word separately, and there can by multiple lexical innovations that spread in parallel), nor that lexical change is inert (the basic reproductive ratio, as a dimensionless quantity, does not measure rate of growth). However, the finding informs us about the relative impact that each individual has on such change phenomena. Importantly, $R_0$ is defined in such a way that it only considers propagation events in which one individual already knows and uses a certain constituent while another individual does not do so. It does not capture events in which both individuals already know a word and where the usage of the word is promoted in

one individual as a reaction to the other one using it, say, through psychological priming effects or socio-pragmatic factors like conformism and prestige [82, 88, 89]. See [90] for a socio-pragmatically more elaborated version of the model including conformity and non-conformity biases.

On a more general level, it is evident that the model, if taken for granted, together with the three ways for deriving the basic reproductive ratio provides links between three different domains: (i) lexical acquisition, (ii) synchronic lexical distribution, and (iii) lexical change. In particular, it provides a mechanistic link relevant to three lines of research. First, the link between language acquisition and change was subject to multiple studies in the field of evolutionary linguistics [91]. Generally speaking, words that are learned early are diachronically more stable in that they are less likely to change their form [28, 71, 92] or meaning [93]. One mechanism that was suggested to explain this link is that of cognitive entrenchment [94–96]. Words that are learned early display a higher degree of routinization (e.g., in their production, retrieval, or perception), remain stable in linguistic usage, and are therefore less likely to get ousted. In fact, the epidemiological model discussed in this contribution is agnostic with respect to such mechanisms: it predicts that early acquired words are diachronically more stable simply because early acquisition increases the length of the prolific (i.e., infectious) period in individuals, thus promoting successful propagation. In that sense, the model can also function as a–let us call it: population dynamic–base line against which hypotheses about the effects of cognitive entrenchment need to be tested.

Second, and connected to this, the basic reproductive ratio links synchronic lexical distribution (prevalence) and diachronic lexical change (growth). In short, the prediction is that what is contemporarily prevalent has been successfully growing in the past. While this might sound trivial at first sight, the mathematical intricacies involved in that link provide more information. The idea is that a word's prevalence *at its population dynamic equilibrium* (i.e., neither when it is growing nor declining) entails how fast it has been growing in the past, and *vice versa*. Quantitative research on diachronic lexical change largely draws on investigating frequency trajectories, or (trajectories of) dispersion measures based on token frequency and document frequency [97–99]. As discussed before, it is not trivial to derive lexical prevalence from corpus data [51], so that using a combination of frequency trajectories and contemporary prevalence estimates (as in this study) or conducting apparent-time diachronic analyses [100, 101] (Boberg 2004; Bailey 2004) remain the only feasible options. Both approaches come with their own problems. Most notably, they can only be reasonably applied to recent phenomena (words that have been prevalent a century ago but got ousted afterwards cannot be captured by contemporary surveys). Likewise, both approaches are challenged by speaker internal changes [102]. We conclude that more research is needed to reliably test the link between equilibrium prevalence and growth suggested by the model.

Third, and finally, the basic reproductive ratio links acquisition with synchronic lexical distributions: words that are acquired early are used by a large share of the population [48], a link that was subject to research on language learning. It was shown that learning from multiple informants via distributed exposure promotes acquisition, both in infants [103, 104] and adults [105]. Again, the epidemiological model does not necessarily need to rely on mechanisms like this. Even if all individuals learn words at a constant rate, independent of their age, it trivially predicts that early acquired words are more widely spread and *vice versa* because (a) early acquisition increases the prolific period and hence the number of users of a word and (b) because words are more quickly adopted by a non-users if there are many users around. What is potentially more interesting than the correlation between ease of acquisition and prevalence is how exactly they are linked to each other through the basic reproductive ratio (more specifically: $1/(1-p_U) = R_0 = LE/AoA$ and hence $p_U = 1-AoA/LE$): we have seen that straightforward

transformations of age of acquisition and prevalence lead to estimates of the basic reproductive ratio that fall into similar ranges. Crucially, these transformations are theoretically motivated and follow from relatively basic epidemiological considerations (as opposed to assumptions about how age, learning, and cognition hang together). That is, the basic reproductive ratio functions as a theoretically grounded way of normalizing data from diverse domains to map them onto the same scale.

## 5. Conclusion

In this contribution, we have explored ways of estimating the basic reproductive ratio in the linguistic context [25]. For this, we have revisited a relatively simple population dynamic model of the spread of single words, originally established in the field of mathematical epidemiology. Under the assumption that the model roughly captures how words spread through speaker populations, our study allows for the following conclusions. First, estimates derived from diachronic frequency trajectories only fit to those obtained from prevalence and age of acquisition when words do not change substantially in frequency. Even in that case, however, the correlation is far from strong. Second, and more optimistically, judging from the good fit of the estimates derived from prevalence and age of acquisition, it can be concluded that one does not necessarily need knowledge of the past development of words in order to determine the basic reproductive ratio.

Our analysis shows that periphery words that have emerged relatively recently, on average, show a basic reproductive ratio of about $R_0 = 6$. This not only lets us put numbers on how stable lexical innovations are (viz. the herd-immunity threshold of about 83%), but it also provides insight into the question of how often successful propagation of a single lexical innovation among individuals takes place at all. The answer is: only a handful of such events during the whole lifetime. Arguably, assertions like this cannot be derived by investigating token frequencies drawn from diachronic corpus data. Hence, we think that epidemiological models when they are applied to linguistics [3, 25, 26, 40], despite their high degree of abstraction, can provide interesting insights into how language evolves.

## Acknowledgments

I would like to thank Nikolaus Ritt and Quentin Feltgen for their invaluably helpful comments on earlier versions of this manuscript, as well as Richard Blythe and a second anonymous reviewer for their huge amount of comments and questions, all of which have helped considerably to improve the research, its presentation, and discussion.

## Author Contributions

**Conceptualization:** Andreas Baumann.

**Data curation:** Andreas Baumann.

**Formal analysis:** Andreas Baumann.

**Funding acquisition:** Andreas Baumann.

**Investigation:** Andreas Baumann.

**Methodology:** Andreas Baumann.

**Project administration:** Andreas Baumann.

**Resources:** Andreas Baumann.

**Software:** Andreas Baumann.

**Supervision:** Andreas Baumann.

**Validation:** Andreas Baumann.

**Visualization:** Andreas Baumann.

**Writing – original draft:** Andreas Baumann.

**Writing – review & editing:** Andreas Baumann.

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
