## [Decision Letter · Decision Letter 0]

1 Aug 2024

PONE-D-24-25073Lexical innovations are rarely passed on during one’s lifetime: epidemiological perspectives on estimating the basic reproductive ratio of wordsPLOS ONE

Dear Dr. Baumann,

Thank you for submitting your manuscript to PLOS ONE. After careful consideration, we feel that it has merit but does not fully meet PLOS ONE’s publication criteria as it currently stands. Therefore, we invite you to submit a revised version of the manuscript that addresses the points raised during the review process.

Please see comments from myself and two reviewers below. Please submit your revised manuscript by Sep 15 2024 11:59PM. If you will need more time than this to complete your revisions, please reply to this message or contact the journal office at plosone@plos.org. Please include the following items when submitting your revised manuscript:A rebuttal letter that responds to each point raised by the academic editor and reviewer(s). You should upload this letter as a separate file labeled 'Response to Reviewers'.A marked-up copy of your manuscript that highlights changes made to the original version. You should upload this as a separate file labeled 'Revised Manuscript with Track Changes'.An unmarked version of your revised paper without tracked changes. You should upload this as a separate file labeled 'Manuscript'.

We look forward to receiving your revised manuscript.

Kind regards,

Søren Wichmann, PhD

Academic Editor

PLOS ONE

Journal Requirements:

Additional Editor Comments:

The two reviewers raise many questions, which you should try to take into account to all possible extent. In a few cases a bit more is asked for than is possible to deliver in terms of data available so the best response in these cases would be to briefly indicate the limitations and perhaps (again briefly) speculating on how results might be different given more information. One of such "what if" questions of my own relates to how things would look if we know more about differences in the amount of contacts among individuals. The estimated R0 looks pretty high for an average among all individuals, but one would expect that there is a non-normal distribution of the amount of contact individuals have when contacts include readers/listeners of printed and audio media, so an estimated mean might not be very meaningful. Perhaps the distribution of contacts is something like a power law, with the vast majority of individuals belonging to a fat tail contributing even less to the spread of innovations than the R0 = 6 would suggest. In whatever way this is done (just mentioning, speculating, trying to find data...), I think it needs to be taken into account to at least some minimal extent that lexical diffusion is different from the spread of diseases in that a single individual can reach a large part of a population in one instance.

There are a few typos etc.:

l. 90: an average -> on average

l. 91-92: perhaps move "not yet knowing the word" up to just after "contacts"

l. 96-97: Heffernan et al. (2005) -> the paper of Heffernan et al. (23)

l. 118: see van den Driessche (2017) -> see (24)

l. 524: the lexicon growths -> the lexicon grows

l. 530: data, mostly -> data mostly

l. 615: delete "(Boberg 2004; Bailey 2004)"

Reviewers' comments:

Reviewer's Responses to Questions

**Comments to the Author**

1. Is the manuscript technically sound, and do the data support the conclusions?

Reviewer #1: Partly

Reviewer #2: Yes

2. Has the statistical analysis been performed appropriately and rigorously? 

Reviewer #1: No

Reviewer #2: Yes

3. Have the authors made all data underlying the findings in their manuscript fully available?

Reviewer #1: Yes

Reviewer #2: Yes

4. Is the manuscript presented in an intelligible fashion and written in standard English?

Reviewer #1: Yes

Reviewer #2: Yes

5. Review Comments to the Author

Reviewer #1: The paper proposes an interesting method transfer from epidemiology to linguistics, but suffers from simplification of language data and diachronic change factors to the point that I am not entirely sure it contributes much to linguistics in its current form, where the predictions are not tested yet rather strong claims about language change are made in the end. The unciritical usage of Google Ngrams for lexical data is also questionable. I would suggest the author to either: a) repurpose this as a method proposal paper, acknowledge the limitations, and remove the sweeping claims in the end; or b) figure out a way to actually test these predictions on a test set of lexical change cases, and work out how to estimate a confidence interval around the proposed R values and resulting claims. Don't get me wrong, it is an interesting idea, but the massive simplifications and ignorance of the nature of written corpus data will, I fear, only lead to backlash from linguists. Whereas this has potential to be actually useful, if presented carefully and tested appropriately.

I will comment on a selection of passages below and attempt to explain my reasoning (the passages are copied from the pdf so line breaks and line numbers intersect with text, don't mind that).

After that, we will test predictions against diverse diachronic and synchronic

linguistic data from four languages (English, Spanish, German, Italian; sections 2-3).

--- This is my main problem with the paper, I don't see how the model is really "tested".

If there were, for instance, hypothetical political measures for banning a certain word,

232 one would need to ensure that at least individuals are not able to adopt that word.

--- This does not take into account uneven language policies and societal attitudes, e.g. in earlier centuries the upper and lower classes of these countries differed in their language use, and even today usage of certain words can be restricted to certain groups who may internally or externally regulate (think of racial slurs for example).

We test the predictions outlined above on the lexical level, i.e., by inspecting dynamics of words

--- This is another problem I see, there is zero discussion on what counts as a word when it comes to frequency data - is it any space-separated token in a corpus, a lemma, or something else? There is no discussion on lemmatization, but the word lists used are presumably lemmas, so were only the forms corresponding to dictionary forms counted? E.g. for English nouns it's the singular, so were plurals (analogously other forms of verbs) not counted in frequencies? English is morphologically simple, but e.g. German is already more complex with each lemma having a number of forms. How does that affect results?

words that have increased in frequency during the past two decades,

243 and we will refer to such words as lexical innovations. We do so irrespective of whether these words

244 have already been present initially at low frequency

--- So which is it, innovations (not present before), or words that were low frequency and afterwards higher frequency?

Moreover, we only consider periphery words, which we define,

247 somewhat simplistically, as words that are not used by everyone

--- I am not sure what this even means. The corpora used here have no data on speakers.

250 estimates for a large set of English from Kuperman et al. (42).

Prevalence estimates for English

258 were taken from Brysbaert et al

--- This is my other gripe - how are age of acquisition estimates of modern-day children supposed to say anything about (presumably mostly adult) lexical innovations of literate language users in 1820? What is the connection here?

Note that the accuracy of the prevalence estimates substantially depends on sample size, i.e., on

270 the number of participants they are based on. The respective sample sizes are about 300 for English,

271 about 23 for German, about 50 for Spanish, and 25 for Italian

--- Am I to understand correctly that these sweeping claims about how lexical innovations work in human language are based not only on modern estimates but on a handful of e.g. 23 speaker optinions per language?

For German, Spanish, and Italian, we needed

280 to resort to normalized frequency trajectories extracted from Google books

--- Google Books Ngrams is a highly questionable source of language information - with some reservations it can be useful for e.g. grammatical things, but it has been shown to be questionable for lexical research purposes, as its sampling is pretty random and what may appear as "innovations" can easily be just sampling noise. Please see "Characterizing the Google Books Corpus: Strong Limits to Inferences of Socio-Cultural and Linguistic Evolution" by Pechenick et al and "The Impact of Lacking Metadata for the Measurement of Cultural and Linguistic Change Using the Google Ngram Data Sets—Reconstructing the Composition of the German Corpus in Times of WWII" by Koplenig, and consider if this is a valid corpus for your study.

Also, think for a moment about the speaker communities. You are comparing US English where the speaker population went from around 8M to 300M during the COHA corpus period; Italian which is also spoken mostly in one country, German spoken in 4+ countries, and Spanish which by now has half a billion native speakers and is the official language in 20+ countries (in varieties which the Google Books ngram data does not differentiate between). How are these even remotely comparable in a model that relies so much on the individual speaker, and how can such a model reliably applied to such volatile and heterogenous populations?

We use the collected age-of-acquisition estimates, assume a life expectancy of 75 years

(https://data.worldbank.org/), and estimate the basic reproductive ratio as (i) 0

 300 = /. Note

301 that life expectancy has increased through the past two centuries and that women have, on average, a

302 higher life expectation than men, so that the assumption of a single point estimate is clearly simplistic.

--- Life expectancy in 1850 in the US was around 40 years, almost half your assumption. I don't see why this simplification is made, instead of working known expectancy estimates into your model.

303 In the UK, for instance, life expectancy has increased from about 71 in the early 1960s to about 80 in

304 the 2000s (Fig 3, left). Notably, the UK displays the smallest increase in life expectancy.

--- The COHA is US American English. I don't see how the UK is relevant here.

It is not at

328 all easy to obtain word prevalence based on corpus data. This is because corpora often feature only a

329 small number of authors and because the amount of text per author is likely not representative of their

330 linguistic knowledge (let alone the fact that the author population in linguistic corpora is not

331 necessarily representative of the whole population

--- This is a massive problem on its own, acknowledged but not taken into account - but the related problem is differential number of speakers over time - more people simply became literate over 200 years in these countries, so if the corpus is randomly sampled, the number of speakers in it should be significantly different, and ignoring this surely introduces bias in a model so heavily reliant on the unit of individual.

Just a note on 3D plots, and tables and figures with abbreviations - these are just very hard to read. Would be much easier if the terms were written out on the axes and in tables. 3D plots like these are almost unreadable; could just use color for the 3rd dimension where needed.

382 The analysis of expected values for the basic reproductive ratio (for all languages and methods) shows

383 that estimates typically range from about 3 to about 14

--- Here's my third main issue with the paper: what do we learn from this, and how do we know if these numbers are anywhere near true values? There is some post-hoc discussion later, but what use is this number for linguistics and advancing our understanding of how language works and changes?

545 Given the limitations discussed in the previous section, does the approach adopted here represent an

546 accurate model of lexical spread? Certainly not – as it depends on simplistic assumptions. If the model

547 assumptions are met, however, our analysis suggests that periphery words, on average, show a basic

548 reproductive ratio of about 0 ≅ 6, with inter-quartile ranges going from about 2 to 17, depending on

549 the language and estimation method. This puts spreading periphery words – in certain settings –

550 roughly on par with infectious diseases like Rubella

--- Again, rhetorically, so what? The number that may or may not be correct is like a number if a completely unrelated domain. What is the confidence/margin of error here, given all the (considerable) limitations here pertaining to corpus data, population data and the model, are we talking 6±1 or 6±6?

571 Finally, the estimate 0 ≅ 6 also tells us about the role that individuals play in the propagation of new

572 words through contact networks. Mean personal network size was shown to equal about 600 (80). This

573 reflects all individuals that one gets to know during one’s life time. This means that a single individual

574 that picks up a lexical innovation passes it on to only about 1% of their personal contacts – notably,

--- Exactly the same critique as above.

575 throughout the whole period during which that individual knows and uses that word. If one considers

576 an average offspring count of about 1 (per individual),

--- This is below rate of replacement, and well below the average number of children historically. Making this assumption equates to assuming these four languages or their speaker populations have gone extinct by now.

As a final comment, if such a model would be tested against known case, I expect there would need to be some way to account for an error term or variance left undescribed. These current estimates assume a singular process of lexical innovation, whereas language change is complex and affected by numerous factors mostly or entirely ignored here. To provide some ideas:

- Age and mixing is briefly discussed in the end, but nothing about sociolinguistic prestige (tons of literature on that)

- Selection and drift literature is cited but it seems like drift could be one way to conceptualize the residual/error not described by this model (for a more recent update since the Newberry paper cited here, see e.g. "Reliable identification of selection mechanisms in language change").

- Borrowings are mentioned, but how the proportion of L2 speakers affects language is ignored (see e.g. "Larger communities create more systematic languages", "Language Structure Is Partly Determined by Social Structure")

- Stadler et al is cited but this potential mechanism is ignored; relatedly, other works have looked into neighborhood/topical effects ("Quantifying the dynamics of topical fluctuations in language"

"Where New Words Are Born: Distributional Semantic Analysis of Neologisms and Their Semantic Neighborhoods")

- The issue of semantic change is practically ignored - it is not just new forms being innovated, but new meanings arise and old words are given new meanings (maybe useful: "Algorithms in the historical emergence of word senses", "Survey of Computational Approaches to Lexical Semantic Change")

I am not saying a model would need to necessarily include parameters for all these things, but some acknowledgement that it is not just neutral disease-like passing of new words would be prudent, and if the model were to be evaluated on some test set, it would make sense to reserve an error term or otherwise account for the fact that there are other things happening in language change.

Reviewer #2: In this work, the author views lexical dynamics as an infection process, whereby a particular lexical item can infect a user and thus spread through a population. In particular they focus on computing the basic reproduction number R0 for four different languages through three different methods. Subject to underlying assumptions on the dynamics being correct, all three methods should give similar results and the author finds that they do indeed correlate well, with a median value (over words in language) of around 6. The author performs a careful statistical study of the results, showing that the correlations in particular, and the findings in general, are robust. The discussion section does a good job of examining the implications of the results for language dynamics, and in addressing the limitations of the study.

Overall I have little to criticise, although I think the manuscript could benefit from a little more discussion of the following points:

1. The comment that "most average R0 estimates fall into a similar range, across languages ... is reassuring ... and potentially more revealing" deserves expansion. The easiest R0 to intuit is the the one obtained by dividing life expectancy by age-of-acquisition. A value of 6 is therefore indicative of the fact that the median word is typically acquired at around 1/6th of one's overall lifetime, which seems broadly reasonable even given historical changes in life expectancy. The question then is: how much scope is there for variation in values of R0 given how they are defined? If the answer is 'not much' then we should perhaps not be too surprised that we get similar ballpark figures back. (I might argue that the correlations between values from the different methods are therefore more revealing, because there it's less obvious that the variables that they depend on should be related.)

2. Following from point 1, it would help to flag up the basic assumptions behind the model defined by the equation on p8. (NB: it is very helpful to number equations, as it makes it easier for readers to refer to them.) Key among these are the chosen number of compartments, and the fact that populations are assumed to be well-mixed. These assumptions are discussed towards the end of the manuscript, but I think it is worthwhile to emphasise that the different ways one can define R0 are a consequence of these assumptions. Thus if one does not get similar values from the different methods, one possible explanation is that the underlying assumptions are not valid.

3. Related to point 2, it would help to clarify what the infection event that causes a transition from the N to U compartments is envisaged to be and what the characteristic timescale of infection is. Within the 'age of acquisition' approach, it seems implicitly the case that N to U event is a childhood language acquisition event, which one could potentially question for lexical dynamics. For example, when there is an innovation, a word could also spread through an adult population without having to wait to be learnt by the next generation of children. I think this impacts on the discussion around the relatively small number of people that we infect with our lexicon: if we have only childhood language acquisition in mind, then 6 is actually quite a big number, since most individuals tend to act as a caregiver to a small number of children. The timescale question meanwhile I think is important when making comparisons to diseases like Rubella and Covid-19. R0 doesn't say anything about timescales - just how many people you pass the infection onto, on average - but it is clear from these comparisons that the despite having different R0, the timescale over which a lexical item infects a population is much longer than that of an infectious disease. It is not immediately obvious to me what sets the timescale in the linguistic case, especially if infection can happen through use as well as acquisition. It would be interesting if the authors could add any insights they have on this issue to the manuscript.

6. PLOS authors have the option to publish the peer review history of their article (what does this mean?). If published, this will include your full peer review and any attached files.

Reviewer #1: No

Reviewer #2: **Yes: **Richard Blythe

---

## [Author Response · Author response to Decision Letter 0]

26 Sep 2024

All responses can be found in the 'Response to reviewers' document attached at the bottom.

---

## [Editor Report · Decision Letter 1]

7 Oct 2024

Lexical innovations are rarely passed on during one’s lifetime: epidemiological perspectives on estimating the basic reproductive ratio of words

PONE-D-24-25073R1

Dear Dr. Baumann,

We’re pleased to inform you that your manuscript has been judged scientifically suitable for publication and will be formally accepted for publication once it meets all outstanding technical requirements.

Kind regards,

Søren Wichmann, PhD

Academic Editor

PLOS ONE
---

## [Editor Report · Acceptance letter]

18 Oct 2024

PONE-D-24-25073R1 

PLOS ONE

Dear Dr. Baumann, 

I'm pleased to inform you that your manuscript has been deemed suitable for publication in PLOS ONE. Congratulations! Your manuscript is now being handed over to our production team.

Kind regards, 

on behalf of

Dr. Søren Wichmann 

Academic Editor

PLOS ONE